# Language Models Encode Collaborative Signals in Recommendation

## Abstract

Recent studies empirically indicate that language models (LMs) encode rich world knowledge beyond mere semantics, attracting significant attention across various fields. However, in the recommendation domain, it remains uncertain whether LMs implicitly encode user preference information. Contrary to the prevailing understanding that LMs and traditional recommender models learn two distinct representation spaces due to a huge gap in language and behavior modeling objectives, this work rethinks such understanding and explores extracting a recommendation space directly from the language representation space. Surprisingly, our findings demonstrate that item representations, when linearly mapped from advanced LM representations, yield superior recommendation performance. This outcome suggests a homomorphic relationship between the language representation space and an effective recommendation space, implying that collaborative signals may indeed be encoded within advanced LMs. Motivated by these findings, we propose a simple yet effective collaborative filtering (CF) model named **AlphaRec**, which utilizes language representations of item textual metadata (*e.g.,* titles) instead of traditional ID-based embeddings. Specifically, AlphaRec is comprised of three main components: a multilayer perceptron (MLP), graph convolution, and contrastive learning (CL) loss function, making it extremely easy to implement and train. Our empirical results show that AlphaRec outperforms leading ID-based CF models on multiple datasets, marking the first instance of such a recommender with text embeddings achieving this level of performance. Moreover, AlphaRec introduces a new text-based CF paradigm with several desirable advantages: being easy to implement, lightweight, rapid convergence, superior zero-shot recommendation abilities in new domains, and being aware of user intention.

## 1 Introduction

Language models (LMs) have achieved great success across various domains [3–7], prompting a critical question about the knowledge encoded within their representation spaces. Recent studies empirically find that LMs extend beyond semantic understanding to encode comprehensive world knowledge about various domains, including game states [8], lexical attributes [9], and even concepts of space and time [10] through language modeling. However, in the domain of recommendation where the integration of LMs is attracting widespread interest [11–15], it remains unclear whether LMs inherently encode relevant information on user preferences and behaviors. One possible reason is the significant difference between the objectives of language modeling for LMs and user behavior modeling for recommenders [16–19].

Currently, one prevailing understanding holds that general LMs and traditional recommenders encode two distinct representation spaces: the language space and the recommendation space (*i.e.,* user and item representation space), each offering potential enhancements to the other for

Submitted to 38th Conference on Neural Information Processing Systems (NeurIPS 2024). Do not distribute.

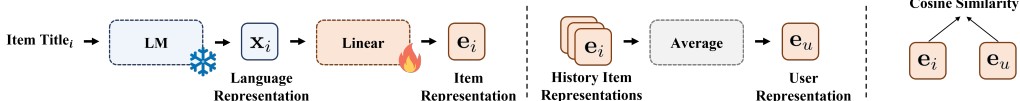

(a) Linearly mapping language representations into the recommendation space

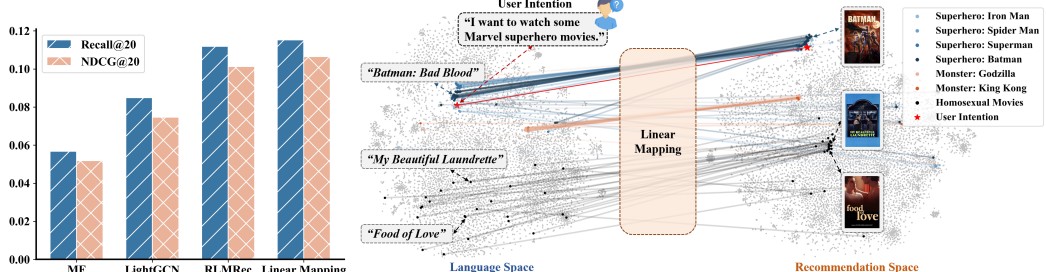

(b) Performance comparison    (c) The t-SNE representations of movies and user intention in two spaces.

Figure 1: Linearly mapping item titles in language representation space into recommendation space yields superior recommendation performance on Movies & TV [1] dataset. (1a) The framework of linear mapping. (1b) The recommendation performance comparison between leading CF recommenders and linear mapping. (1c) The t-SNE [2] visualizations of movie representations, with colored lines linking identical movies or user intention across language space (left) and linearly projected recommendation space (right).

recommendation tasks [17, 20]. On the one hand, when using LMs as recommenders, aligning the language space with the recommendation space could significantly improve the performance of LM-based recommendation [14, 21–23]. Various alignment strategies are proposed, including fine-tuning LMs with recommendation data [15, 16, 24–26], incorporating embeddings from traditional recommenders as a new modality of LMs [17, 20, 27], and extending the vocabulary of LMs with item tokens [18, 19, 28–31]. On the other hand, when using LMs as the enhancer, traditional recommenders greatly benefit from from leveraging text representations [32–45], semantic and reasoning information [46–49], and generated user behaviors [50, 51]. Despite these efforts, explicit explorations of the relationship between language and recommendation spaces remain largely unexplored.

In this work, we rethink the prevailing understanding and explore whether LMs inherently encode user preferences through language modeling. Specifically, we test the possibility of directly deriving a recommendation space from the language representation space, assessing whether the representations of item textual metadata (*e.g.,* titles) obtained from LMs can independently achieve satisfactory recommendation performance. Positive results would imply that user behavioral patterns, such as **collaborative signals** (*i.e.,* user preference similarities between items) [52, 53], may be implicitly encoded by LMs. To test this hypothesis, we employ linear mapping to project the language representations of item titles into a recommendation space (see Figure 1a). Our observations include:

- Surprisingly, this simple linear mapping yields high-quality item representations, which achieve exceptional recommendation performance (see Figure 1b and experimental results in Section 2).

- The clustering of items is generally preserved from the language space to the recommendation space (see Figure 1c). For example, movies with the theme of superheroes and monsters are gathering in both language and recommendation spaces.

- Interestingly, the linear mapping effectively reveals preference similarities that may be implicit or even obscure in the language space. For instance, while certain movies, such as those of homosexual movies (illustrated in Figure 1c), show dispersed representations in the language space, their projections through linear mapping tend to cluster together, reflecting their genres affiliation.

These findings indicate a homomorphic relationship between the language representation space of LMs and an effective item representation space for recommendation. Motivated by this insight, we propose a new text-based recommendation paradigm for general collaborative filtering (CF), which utilizes the pre-trained language representations of item titles as the item input and the average historical interactions' representations as the user input. Different from traditional ID-based CF models [54, 55, 52] that heavily rely on trainable user and item IDs, this paradigm solely uses

pre-trained LM embeddings and completely abandons ID-based embeddings. In this paper, to fully explore the potential of advanced language representations, we adopt a simple model architecture consisting of a two-layer MLP with graph convolution, and the popular contrastive loss, InfoNCE [56–58], as the objective function. This model is named **AlphaRec** for its originality and a series of good properties.

Benefiting from paradigm shifts from ID-based embeddings to language representations, AlphaRec presents three desirable advantages. First, AlphaRec is notable for its simplicity, lightweight, rapid convergence, and exceptional recommendation performance (see Section 4.1). We empirically demonstrate that, for the first time, such a simple model with embeddings from pre-trained LMs can outperform leading CF models on multiple datasets. This finding strongly supports the possibility for developing language-representation-based recommender systems. Second, AlphaRec exhibits a strong zero-shot recommendation capability across untrained domains (see Section 4.2). By co-training on three Amazon datasets (Books, Movies & TV, and Video Games) [1], AlphaRec can achieve performance comparable to the fully-trained LightGCN on entirely different platforms (MovieLens-1M [59] and BookCrossing [60]), and even exceed LightGCN in a completely new domain (Amazon Industrial), without additional training on these target datasets. This capability underscores AlphaRec's potential to develop more general recommenders. Third, AlphaRec is user-friendly, offering a new research paradigm that enhances recommendation by leveraging language-based user feedback (see Section 4.3). Endowed with its inherent semantic comprehension of language representations, AlphaRec can refine recommendations based on user intentions expressed in natural language, enabling traditional CF recommenders to evolve into intention-aware systems through a straightforward paradigm shift.

## 2 Uncovering Collaborative Signals in LMs via Linear Mapping

In this section, we aim to explore whether LMs implicitly encode collaborative signals in their representation spaces. We first formulate the personalized item recommendation task, then detail the linear mapping and its empirical findings. Empirical evidence indicates a homomorphic relationship between the representation spaces of advanced LMs and effective recommendation spaces.

**Task formulation.** Personalized item recommendation with implicit feedback aims to select items $i \in \mathcal{I}$ that best match user $u$'s preferences based on binary interaction data $\mathbf{Y} = [y_{ui}]$, where $y_{ui} = 1$ ($y_{ui} = 0$) indicates user $u \in \mathcal{U}$ has (has not) interacted with item $i$ [58]. The primary objective of recommendation is to model the user-item interaction matrix $\mathbf{Y}$ using a scoring function $\hat{y} : \mathcal{U} \times \mathcal{I} \to \mathbb{R}$, where $\hat{y}_{ui}$ measures $u$'s preference for $i$. The scoring function $\hat{y}_{ui} = s \circ \phi_\theta(\mathbf{x}_u, \mathbf{x}_i)$ comprises three key components: pre-existing features $\mathbf{x}_u$ and $\mathbf{x}_i$ for user $u$ and item $i$, a representation learning module $\phi_\theta(\cdot, \cdot)$ parametrized by $\theta$, and a similarity function $s(\cdot, \cdot)$. The representation learning module $\phi_\theta$ transfers $u$ and $i$ into representations $\mathbf{e}_u$ and $\mathbf{e}_i$ for similarity matching $s(\mathbf{e}_u, \mathbf{e}_i)$, and the Top-$K$ highest scoring items are recommended to $u$.

Different recommenders employ various pre-existing features $\mathbf{x}_u, \mathbf{x}_i$ and representation learning architecture $\phi_\theta(\cdot, \cdot)$. Traditional ID-based recommenders use one-hot vectors as pre-existing features $\mathbf{x}_u, \mathbf{x}_i$. The choice of ID-based representation learning architecture $\phi_\theta$ can vary widely, including ID-based embedding matrix [54], multilayer perception [61], graph neural network [52, 62], and variational autoencoder [63]. The commonly used similarity function is cosine similarity [64, 57] $s(\mathbf{e}_u, \mathbf{e}_i) = \frac{\mathbf{e}_u{}^\top \mathbf{e}_i}{\|\mathbf{e}_u\| \cdot \|\mathbf{e}_i\|}$, which we adopt in this paper.

**Linear mapping.** Building on the extensive knowledge encoded by LMs, we explore utilizing LMs as feature extractors, leveraging the language representations of item titles as initial item feature $\mathbf{x}_i$. For initial user feature $\mathbf{x}_u$, we use the average of the title representations of historically interacted items, defined as $\mathbf{x}_u = \frac{1}{|\mathcal{N}_u|} \sum_{i \in \mathcal{N}_u} \mathbf{x}_i$, where $\mathcal{N}_u$ is the set of items user $u$ has interacted with. Detailed procedures for obtaining these language-based features are provided in Appendix B.2. We select a trainable linear mapping matrix $\mathbf{W}$ as the representation learning module $\phi_\theta$, setting $\mathbf{e}_u = \mathbf{W}\mathbf{x}_u$ and $\mathbf{e}_i = \mathbf{W}\mathbf{x}_i$. To learn the linear mapping $\mathbf{W}$, we adopt the InfoNCE loss [56] as the objective function, which has demonstrated state-of-the-art performance in both ID-based [65, 66] and LM-enhanced collaborative filtering (CF) recommendations [47] (refer to Equation (4) for the formula). The overall framework of the linear mapping process is illustrated in Figure 1a. We directly use linearly mapped representations $\mathbf{e}_u$ and $\mathbf{e}_i$ to calculate the user-item similarity $s(\mathbf{e}_u, \mathbf{e}_i)$ for recommendation. High performance on the test set would suggest that collaborative signals (*i.e.,* user

Table 1: The recommendation performance of linear mapping comparing with classical CF baselines.

| | Books | | | Movies & TV | | | Video Games | | |
|---|---|---|---|---|---|---|---|---|---|
| | Recall | NDCG | HR | Recall | NDCG | HR | Recall | NDCG | HR |
| MF (Rendle et al., 2012) | 0.0437 | 0.0391 | 0.2476 | 0.0568 | 0.0519 | 0.3377 | 0.0323 | 0.0195 | 0.0864 |
| MultVAE (Liang et al., 2018) | 0.0722 | 0.0597 | 0.3418 | 0.0853 | 0.0776 | 0.4434 | 0.0908 | 0.0531 | 0.2211 |
| LightGCN (He et al., 2021) | 0.0723 | 0.0608 | **0.3489** | 0.0849 | 0.0747 | 0.4397 | 0.1007 | 0.0590 | 0.2281 |
| Linear Mapping | | | | | | | | | |
| **BERT** | 0.0226 | 0.0194 | 0.1240 | 0.0415 | 0.0399 | 0.2362 | 0.0524 | 0.0309 | 0.1245 |
| **RoBERTa** | 0.0247 | 0.0209 | 0.1262 | 0.0406 | 0.0387 | 0.2277 | 0.0578 | 0.0338 | 0.1339 |
| **Llama2-7B** | 0.0662 | 0.0559 | 0.3176 | 0.1027 | 0.0955 | 0.4952 | 0.1249 | 0.0729 | 0.2746 |
| **Mistral-7B** | 0.0650 | 0.0544 | 0.3124 | 0.1039 | 0.0963 | 0.4994 | 0.1270 | 0.0687 | 0.2428 |
| **text-embedding-ada-v2** | 0.0515 | 0.0436 | 0.2570 | 0.0926 | 0.0874 | 0.4563 | 0.1176 | 0.0683 | 0.2579 |
| **text-embeddings-3-large** | 0.0735 | 0.0608 | 0.3355 | 0.1109 | 0.1023 | 0.5200 | 0.1367 | **0.0793** | **0.2928** |
| **SFR-Embedding-Mistral** | **0.0738** | **0.0610** | 0.3371 | **0.1152** | **0.1065** | **0.5327** | **0.1370** | 0.0787 | 0.2927 |

preference similarities between items) have been implicitly encoded in the language representation space [67, 10].

**Empirical findings.** We compare the recommendation performance of the linear mapping method with three classical CF baselines, matrix factorization (MF) [54, 68], MultVAE [63], and LightGCN [55] (see more details about baselines in Appendix C.2.1). We report three widely used metrics Hit Ratio (HR@$K$), Recall@$K$, Normalized Discounted Cumulative Gain (NDCG@$K$)) to evaluate the effectiveness of linear mapping, with $K$ set by default at 20. We evaluate a wide range of LMs, including BERT-style models [4, 5], decoder-only language models [6, 69], and LM-based text embedding models [70, 71] (see Appendix B.1 for details about used LMs).

Table 1 reports the recommendation performance yielded by the linear mapping on three Amazon datasets [1], comparing with classic CF baselines. We observe that the performance of most advanced text embedding models (*e.g.,* text-embeddings-3-large [70] and SFR-Embedding-Mistral [71]) exceed LightGCN on all datasets. We further empirically prove that these improvements do not merely come from the better feature encoding ability (refer to Appendix B.3). These findings indicate the homomorphic relationship between the language representation space of advanced LMs and an effective item representation space for recommendation. Moreover, with the advances in LMs, the performance of item representation linearly mapped from LMs exhibits a rising trend, gradually surpassing traditional ID-based CF models. Representations from early BERT-style models (*e.g.,* BERT [4] and RoBERTa [5]) only show weaker or equal capabilities compared with MF, while the performance of decoder-only LMs (*e.g.,* Llama-7B [6] ) start to match MultVAE and LightGCN.

# 3 AlphaRec

This finding of space homomorphic relationship sheds light on building advanced CF models purely based on LM representations without introducing ID-based embeddings. To be specific, we try to incorporate only three simple components (*i.e.,* nonlinear projection [61], graph convolution [55] and contrastive learning (CL) objectives [56]), to develop a simple yet effective CF model called AlphaRec. It is important to highlight that our approach is centered on exploring the potential of LM representations for CF by integrating essential components from leading CF models, rather than deliberately inventing new CF mechanisms. We present the model structure of AlphaRec in Section 3.1, and compare AlphaRec with two popular recommendation paradigms in Section 3.2.

## 3.1 Method

We present how AlphaRec is designed and trained. Generally, the representation learning architecture $\phi_\theta(\cdot, \cdot)$ of AlphaRec is simple, which only contains a two-layer MLP and the basic graph convolution operation, with language representations as the input features $\mathbf{x}_u, \mathbf{x}_i$. The cosine similarity is used as the similarity function $s(\cdot, \cdot)$, and the contrastive loss InfoNCE [56, 57] is adopted for optimization. For simplicity, we consistently adopt text-embeddings-3-large [70] as the language representation model, for its excellent language understanding and representation capabilities.

**Nonlinear projection.** In AlphaRec, we substitute the linear mapping matrix delineated in Section 2 with a nonlinear MLP. This conversion from linear to nonlinear is non-trivial, for the paradigm shift from ID-based embeddings to LM representations, since nonlinear transformation helps in excavating more comprehensive collaborative signals from the LM representation space with rich semantics (see

discussions about this in Appendix C.2.3) [61]. Specifically, we project the language representation $\mathbf{x}_i$ of the item title to an item space for recommendation with the two-layer MLP, and obtain user representations as the average of historical items:

$$\mathbf{e}_i^{(0)} = \boldsymbol{W}_2 \, \mathrm{LeakyReLU}\left(\boldsymbol{W}_1 \mathbf{x}_i + \boldsymbol{b}_1\right) + \boldsymbol{b}_2, \quad \mathbf{e}_u^{(0)} = \frac{1}{|\mathcal{N}_u|} \sum_{i \in \mathcal{N}_u} \mathbf{e}_i^{(0)}. \tag{1}$$

**Graph convolution.** Graph neural networks (GNNs) have shown superior effectiveness for recommendation [52, 55], owing to the natural user-item graph structure in recommender systems [72]. In AlphaRec, we employ a minimal graph convolution operation [55] to capture more complicated collaborative signals from high-order connectivity [55, 73, 74, 72] as follows:

$$\mathbf{e}_u^{(k+1)} = \sum_{i \in \mathcal{N}_u} \frac{1}{\sqrt{|\mathcal{N}_u|}\sqrt{|\mathcal{N}_i|}} \mathbf{e}_i^{(k)}, \quad \mathbf{e}_i^{(k+1)} = \sum_{u \in \mathcal{N}_i} \frac{1}{\sqrt{|\mathcal{N}_i|}\sqrt{|\mathcal{N}_u|}} \mathbf{e}_u^{(k)}. \tag{2}$$

The information of connected neighbors is aggregated with a symmetric normalization term $\frac{1}{\sqrt{|\mathcal{N}_u|}\sqrt{|\mathcal{N}_i|}}$. Here $\mathcal{N}_u$ ($\mathcal{N}_i$) denotes the historical item (user) set that user $u$ (item $i$) has interacted with. The features $\mathbf{e}_u^{(0)}$ and $\mathbf{e}_i^{(0)}$ projected from the MLP are used as the input of the first layer. After propagating for $K$ layers, the final representation of a user (item) is obtained as the average of features from each layer:

$$\mathbf{e}_u = \frac{1}{K+1} \sum_{k=0}^{K} \mathbf{e}_u^{(k)}, \quad \mathbf{e}_i = \frac{1}{K+1} \sum_{k=0}^{K} \mathbf{e}_i^{(k)}. \tag{3}$$

**Contrastive learning objective.** The introduction of contrasting learning is another key element for the success of leading CF models. Recent research suggests that the contrast learning objective, rather than data augmentation, plays a more significant role in improving recommendation performance [66, 75, 65]. Therefore, we simply use the contrast learning object InfoNCE [56] as the loss function without any additional data augmentation on the graph [76, 57]. With cosine similarity as the similarity function $s(\mathbf{e}_u, \mathbf{e}_i) = \frac{\mathbf{e}_u^\top \mathbf{e}_i}{\|\mathbf{e}_u\| \cdot \|\mathbf{e}_i\|}$, the InfoNCE loss [56, 76, 77] is written as:

$$\mathcal{L}_{\text{InfoNCE}} = -\sum_{(u,i) \in \mathcal{O}^+} \log \frac{\exp\left(s(u,i)/\tau\right)}{\exp\left(s(u,i)/\tau\right) + \sum_{j \in \mathcal{S}_u} \exp\left(s(u,j)/\tau\right)}. \tag{4}$$

Here, $\tau$ is a hyperparameter called temperature [78], $\mathcal{O}^+ = \{(u,i)|y_{ui} = 1\}$ denoting the observed interactions between users $\mathcal{U}$ and items $\mathcal{I}$. And $\mathcal{S}_u$ is a randomly sampled subset of negative items that user $u$ does not adopt.

## 3.2 Discussion of Recommendation Paradigms

We compare the language-representation-based AlphaRec with two popular recommendation paradigms in Table 2 (see more discussion about related works in Appendix A).

**ID-based recommendation (ID-Rec) [52, 54].** In the traditional ID-based recommendation paradigm, users and items are represented by ID-based learnable embeddings derived from a large number of user interactions. While ID-Rec exhibits excellent recommendation capabilities with low training and inference costs [62, 76], it also has two significant drawbacks. Firstly, these ID-based embeddings learned in specific domains are difficult to transfer to new domains without overlapping users and items [37], thereby hindering zero-shot recommendation capabilities. Additionally, there is a substantial gap between ID-Rec and natural languages [34], which makes ID-based recommenders hard to incorporate language-based user intentions and further refine recommendations accordingly.

**LM-based recommendation (LM-Rec) [15, 16, 24].** Benefitting from the extensive world knowledge and powerful reasoning capabilities of LMs [7, 79], the LM-based recommendation paradigm has gained widespread attention [11, 13]. LM-Rec tends to convert user interaction history into text prompts as input for LMs, utilizing pre-trained or fine-tuned LMs in a text generation pattern to recommend items. LM-Rec demonstrates zero-shot and few-shot abilities and can easily understand language-based user intentions. However, LM-Rec faces significant challenges. Firstly, the LM-based model architecture leads to huge training and inference costs, with real-world deployment difficulties.

Table 2: Comparison of recommendation paradigms

| Recommendation Paradigms | Training Cost | Zero-shot Ability | Intention-aware Ability |
|---|---|---|---|
| ID-based | Low | ✗ | ✗ |
| LLM-based | High | ✔ | ✔ |
| Language-representation-based | Low | ✔ | ✔ |

Additionally, limited by the text generation paradigm, LM-based models tend to perform candidate selection [17] or generate a single next item [24]. It remains difficult for LM-Rec to comprehensively rank the entire item corpus or recommend multiple items that align with user interests.

**Language-representation-based recommendation.** We argue that AlphaRec follows a new CF paradigm, which we term the language-representation-based paradigm. This paradigm replaces the ID-based embeddings in ID-Rec with representations from pre-trained LMs, employing feature encoders to map LM representations directly into the recommendation space. Few early studies lie in this paradigm, including using BERT-style LMs to learn universal sequence representations [37, 44], or adopting the same model architecture as ID-Rec with simple input features replacement [34, 35]. These early explorations, which are mostly based on BERT-style LMs, are usually only applicable in certain specific scenarios, such as the transductive setting with the help of ID-based embeddings [37]. This phenomenon is consistent with our previous findings in Section 2, indicating that BERT-style LMs may fail to effectively encode collaborative signals. We point out that AlphaRec is the first recommender in the language-representation-based paradigm to surpass the traditional ID-based paradigm on multiple tasks, faithfully demonstrating the effectiveness and potential of this paradigm.

## 4  Experiments

In this section, we aim to explore the effectiveness of AlphaRec. Specifically, we are trying to answer the following research questions:

- **RQ1:** How does AlphaRec perform compared with leading ID-based CF methods?

- **RQ2:** Can AlphaRec learn general item representations, and achieve good zero-shot recommendation performance on entirely new datasets?

- **RQ3:** Can AlphaRec capture user intention described in natural language and adjust the recommendation results accordingly?

### 4.1  General Recommendation Performance (RQ1)

**Motivation.** We aim to explore whether the language-representation-based recommendation paradigm can outperform the ID-Rec paradigm. An excellent performance of AlphaRec would shed light on the research line of building representation-based recommenders in the future.

**Baselines.** We only consider ID-based baselines in this section. We ignore LM-based methods due to two practical difficulties: the huge inference cost on datasets with millions of interactions and the task limitation of candidate selection or next item prediction. In addition to classic baselines (*i.e.,* MF, MultVAE, and LightGCN) introduced in section 2, we consider two categories of leading ID-based CF baselines: CL-based CF methods: SGL [80], BC Loss [76], XSimGCL [66] and LM-enhanced methods: KAR [48], RLMRec [47]. See more details about baselines in Appendix C.2.1.

**Results.** Table 3 presents the performance of AlphaRec compared with leading CF baselines. The best-performing methods are bold, while the second-best methods are underlined. Figure 2a and Figure 2b report the training efficiency and ablation results. We observe that:

- **AlphaRec consistently outperforms leading CF baselines by a large margin across all metrics on all datasets.** AlphaRec shows an improvement ranging from 6.79% to 9.75% on Recall@20 compared to the best baseline RLMRec [47]. We further conduct the ablation study to explore the reason for its success (see more ablation results in Appendix C.2.2). As shown in Figure 2b, each component in AlphaRec contributes positively. Specifically, the performance degradation caused by replacing the MLP with a linear weight matrix (w/o MLP) indicates that nonlinear transformations can further extract the implicit collaborative signals encoded in the LM representation space.

Table 3: The performance comparison with ID-based CF baselines. The improvement achieved by AlphaRec is significant ($p$-value $<< 0.05$).

| | Books | | | Movies & TV | | | Video Games | | |
| --- | --- | --- | --- | --- | --- | --- | --- | --- | --- |
| | Recall | NDCG | HR | Recall | NDCG | HR | Recall | NDCG | HR |
| MF (Rendle et al., 2012) | 0.0437 | 0.0391 | 0.2476 | 0.0568 | 0.0519 | 0.3377 | 0.0323 | 0.0195 | 0.0864 |
| MultVAE (Liang et al., 2018) | 0.0722 | 0.0597 | 0.3418 | 0.0853 | 0.0776 | 0.4434 | 0.0908 | 0.0531 | 0.2211 |
| LightGCN (He et al., 2021) | 0.0723 | 0.0608 | 0.3489 | 0.0849 | 0.0747 | 0.4397 | 0.1007 | 0.0590 | 0.2281 |
| SGL (Wu et al., 2021) | 0.0789 | 0.0657 | 0.3734 | 0.0916 | 0.0838 | 0.4680 | 0.1089 | 0.0634 | 0.2449 |
| BC Loss (Zhang et al., 2022) | 0.0915 | 0.0779 | 0.4045 | 0.1039 | 0.0943 | 0.5037 | 0.1145 | 0.0668 | 0.2561 |
| XSimGCL (Yu et al., 2024) | 0.0879 | 0.0745 | 0.3918 | 0.1057 | 0.0984 | 0.5128 | 0.1138 | 0.0662 | 0.2550 |
| KAR (Xi et al., 2023) | 0.0852 | 0.0734 | 0.3834 | 0.1084 | 0.1001 | 0.5134 | 0.1181 | 0.0693 | 0.2571 |
| RLMRec (Ren et al., 2024) | 0.0928 | 0.0774 | 0.4092 | 0.1119 | 0.1013 | 0.5301 | 0.1384 | 0.0809 | 0.2997 |
| **AlphaRec** | **0.0991*** | **0.0828*** | **0.4185*** | **0.1221*** | **0.1144*** | **0.5587*** | **0.1519*** | **0.0894*** | **0.3207*** |
| Imp.% over the best baseline | 6.79% | 5.34% | 2.27% | 9.12% | 10.75% | 5.40% | 9.75% | 10.51% | 7.01% |

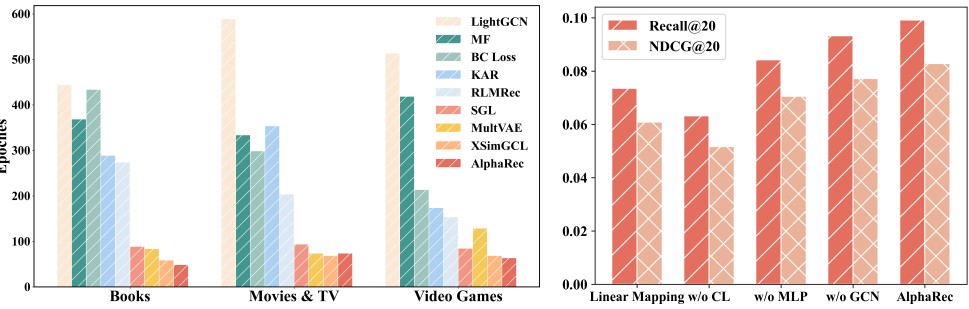

(a) Training efficiency comparison      (b) Ablation study on Books

Figure 2: (2a) The bar charts show the number of epochs needed for each model to converge. AlphaRec tends to exhibit an extremely fast convergence speed. (2b) The effect of each component in AlphaRec on Books dataset.

Moreover, the performance drop from replacing InfoNCE loss [57] with BPR loss [68] (w/o CL) and removing the graph convolution (w/o GCN) suggests that explicitly modeling the collaborative relationships through the loss function and model architecture can further enhance recommendation performance. These findings suggest that, by carefully designing the model to extract collaborative signals, the language-representation-based paradigm can surpass the ID-Rec paradigm.

- **The incorporation of semantic LM representations into traditional ID-based CF methods can lead to significant performance improvements.** We note that two LM-enhanced CF methods, KAR and RLMRec, both show improvements over CL-based CF methods. Nevertheless, the combination of ID-based embeddings and LM representations in these methods does not yield higher results than purely language-representation-based AlphaRec. We attribute this phenomenon to the fact that the performance contribution of these methods mainly comes from the LM representations, which is consistent with the previous findings [34, 44].

- **AlphaRec exhibits fast convergence speed.** We find that the convergence speed of AlphaRec is comparable with, or even surpasses, CL-based methods with data augmentation (*e.g.,* SGL [80] and XSimGCL [66]). Meanwhile, methods based solely on graph convolution (LightGCN [55]) or CL objective (BC Loss [76]) show relatively slow convergence speed, indicating that introducing these modules may not lead to convergence speed improvement. Therefore, we attribute the fast convergence speed of AlphaRec to the homomorphic relationship between the LM representation space and a good recommendation space, so only minor adjustments to the LM representations are needed for recommendation.

## 4.2 Zero-shot Recommendation Performance on Entirely New Datasets (RQ2)

**Motivation.** We aim to explore whether AlphaRec has learned general item representations [37], which enables it to perform well on entirely new datasets without any user and item overlap.

**Task and datasets.** In zero-shot recommendation [38], there is not any item or user overlap between the training set and test set [38, 33], which is different from the research line of cross-domain recommendation in ID-Rec [81]. We jointly train AlphaRec on three source datasets (*i.e.,* Books, Movies & TV, and Video Games), while testing it on three completely new target datasets (*i.e.,*

Table 4: The zero-shot recommendation performance comparison on entirely new datasets. The improvement achieved by AlphaRec is significant (*p*-value << 0.05).

| | | Industrial | | | MovieLens-1M | | | Book Crossing | | |
|---|---|---|---|---|---|---|---|---|---|---|
| | | Recall | NDCG | HR | Recall | NDCG | HR | Recall | NDCG | HR |
| full | MF (Rendle et al., 2012) | 0.0344 | 0.0225 | 0.0521 | 0.1855 | 0.3765 | 0.9634 | 0.0316 | 0.0317 | 0.2382 |
| | MultVAE (Liang et al., 2018) | 0.0751 | 0.0459 | 0.1125 | 0.2039 | 0.3741 | 0.9740 | 0.0736 | 0.0634 | 0.3716 |
| | LightGCN (He et al., 2021) | 0.0785 | 0.0533 | 0.1078 | 0.2019 | 0.4017 | 0.9715 | 0.0630 | 0.0588 | 0.3475 |
| zero-shot | Random | 0.0148 | 0.0061 | 0.0248 | 0.0068 | 0.0185 | 0.2611 | 0.0039 | 0.0036 | 0.0443 |
| | Pop | 0.0216 | 0.0087 | 0.0396 | 0.0253 | 0.0679 | 0.5439 | 0.0119 | 0.0101 | 0.1157 |
| | ZESRec (Ding et al., 2021) | 0.0326 | 0.0272 | 0.0628 | 0.0274 | 0.0787 | 0.5786 | 0.0155 | 0.0143 | 0.1347 |
| | UniSRec (Hou et al., 2022) | 0.0453 | 0.0350 | 0.0863 | 0.0578 | 0.1412 | 0.7135 | 0.0396 | 0.0332 | 0.2454 |
| | **AlphaRec** | **0.0913*** | **0.0573** | **0.1277*** | **0.1486*** | **0.3215*** | **0.9296*** | **0.0660*** | **0.0545*** | **0.3381*** |
| | Imp.% over the best zero-shot baseline | 157.09% | 127.69% | 30.29% | 66.67% | 64.16% | 37.78% | 101.55% | 63.71% | 47.97% |

Movielens-1M [59], Book Crossing [60], and Industrial [1]) without further training on these new datasets. (see more details about how we train AlphaRec on multiple datasets in Appendix C.3.1).

**Baselines.** Due to the lack of zero-shot recommenders in the field of general recommendation, we slightly modify two zero-shot methods in the sequential recommendation [82], ZESRec [37] and UniSRec [37], as baselines. We also incorporate two strategy-based CF methods, Random and Pop (see more details about these baselines in Appendix C.3.2).

**Results.** Table 4 presents the zero-shot recommendation performance comparison on entirely new datasets. The best-performing methods are bold and starred, while the second-best methods are underlined. We observe that:

- **AlphaRec demonstrates strong zero-shot recommendation capabilities, comparable to or even surpassing the fully trained LightGCN.** On datasets from completely different platforms (*e.g.,* MovieLens-1M and Book Crossing), AlphaRec is comparable with the fully trained LightGCN. On the same Amazon platform dataset, Industrial, AlphaRec even surpasses LightGCN, which we attribute to the possibility that AlphaRec implicitly learns unique user behavioral patterns on the Amazon platform [1]. Conversely, ZESRec and UniSRec exhibit a marked performance decrement compared with AlphaRec. We attribute this phenomenon to two aspects. On the one hand, BERT-style LMs [4, 5] used in these works may not have effectively encoded collaborative signals, which is consistent with our findings in Section 2. On the other hand, components designed for the next item prediction task in sequential recommendation [83] may not be suitable for capturing the general preferences of users in CF scenarios.

- **The zero-shot recommendation capability of AlphaRec generally benefits from an increased amount of training data, without harming the performance on source datasets.** As illustrated in Figure 8, the zero-shot performance of AlphaRec, when trained on a mixed dataset, is generally superior to training on one single dataset [37]. Additionally, we also note that training data with themes similar to the target domain contributes more to the zero-shot performance. For instance, the zero-shot capability on MovieLens-1M may primarily stem from Movies & TV. Furthermore, we discover that AlphaRec, when trained jointly on multiple datasets, hardly experiences a performance decline on each source dataset. These findings further point to the general recommendation capability of a single pre-trained AlphaRec across multiple datasets. The above findings also offer a potential research path to achieve general recommendation capabilities, by incorporating more training data with more themes. See more details about these results in Appendix C.3.3.

### 4.3 User Intention Capture Performance (RQ3)

**Motivation.** We aim to investigate whether a straightforward paradigm shift enables pre-trained AlphaRec to perceive text-based user intentions and refine recommendations.

**Task and datasets.** We test the user intention capture ability of AlphaRec on MovieLens-1M and Video Games. In the test set, only one target item remains for each user [84], with one intention query generated by ChatGPT [85, 40] (see the details about how to generate and check these intention queries in Appendix C.4.1). In the training stage, we follow the same procedure as illustrated in Section 2 to train AlphaRec. In the inference stage, we obtain the LM representation $\mathbf{e}_u^{Intention}$ for each user intention query and combine it with the original user representation to get a new user representation as $\tilde{\mathbf{e}}_u^{(0)} = (1-\alpha)\mathbf{e}_u^{(0)} + \alpha\mathbf{e}_u^{Intention}$ [84]. This new user representation is sent into the freezed AlphaRec for recommendation. We report a relatively small $K = 5$ for all metrics to better reflect the intention capture accuracy.

Table 5: The performance comparison in user intention capture.

|  | MovieLens-1M | | Video Games | |
|---|---|---|---|---|
|  | HR@5 | NDCG@5 | HR@5 | NDCG@5 |
| TEM (Bi et al., 2020) | 0.2738 | 0.1973 | 0.2212 | 0.1425 |
| AlphaRec (w/o Intention) | 0.0793 | 0.0498 | 0.0663 | 0.0438 |
| AlphaRec (w Intention) | **0.4704\*** | **0.3738\*** | **0.2569\*** | **0.1862\*** |

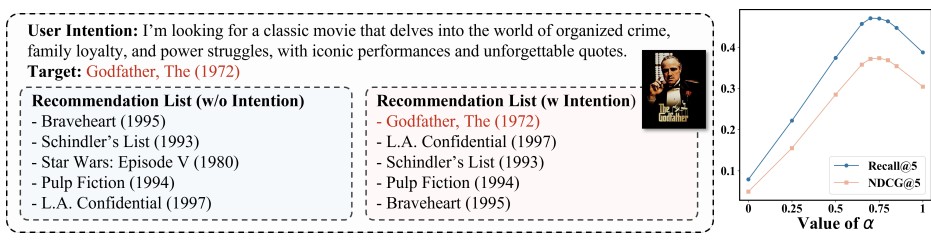

(a) Case study of user intention capture    (b) Effect of $\alpha$

Figure 3: User intention capture experiments on MovieLens-1M. (3a) AlphaRec refines the recommendations according to language-based user intention. (3b) The effect of user intention strength $\alpha$.

**User intention capture results.** Table 5 represents the user intention capture experiment results, compared with the baseline TEM [86]. Clearly, the introduction of user intention (w Intention) significantly refines the recommendations of the pre-trained AlphaRec (w/o Intention). Moreover, AlphaRec outperforms the baseline model TEM by a large margin, even without additional training on search tasks. We further conduct a case study on MovieLens-1M to demonstrate how AlphaRec captures the user (see more case study results in Appendix C.4.3). As shown in Figure 3a, AlphaRec accurately captures the hidden user intention for "Godfather", while keeping most of the recommendation results unchanged. This indicates that AlphaRec captures the user intention and historical interests simultaneously.

**Effect of the intention strength $\alpha$.** By controlling the value of $\alpha$, AlphaRec can provide better recommendation results, with a balance between user historical interests and user intent capture. Figure 3b depicts the effect of $\alpha$. Initially, as $\alpha$ increases, the recommendation performance rises accordingly, indicating that incorporating user intention enables AlphaRec to provide better recommendation results. However, as the $\alpha$ approaches 1, the recommendation performance starts to decrease, which suggests that the user historical interests learned by AlphaRec also play a vital role. The similar effect of $\alpha$ on Video Games is discussed in Appendix C.4.4.

## 5 Limitations

There are several limitations not addressed in this paper. On the one hand, although we have demonstrated the excellence of AlphaRec for multiple tasks on various offline datasets, the effectiveness of online employment remains unclear. On the other hand, although we have successfully explored the potential of language-representation-based recommenders by incorporating essential components in leading CF models, we do not elaboratively focus on designing new components for CF models.

## 6 Conclusion

In this paper, we explored what knowledge about recommendations has been encoded in the LM representation space. Specifically, we found that the advanced LMs representation space exhibits a homomorphic relationship with an effective recommendation space. Based on this finding, we developed a simple yet effective CF model called AlphaRec, which exhibits good recommendation performance with zero-shot recommendation and user intent capture ability. We pointed out that AlphaRec follows a new recommendation paradigm, language-representation-based recommendation, which uses language representations from LMs to represent users and items and completely abandons ID-based embeddings. We believed that AlphaRec is an important stepping stone towards building general recommenders in the future.[1]

---

[1]The broader impact of AlphaRec will be detailed in Appendix E

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

# A Related Works

**Representations in LMs.** The impressive capabilities demonstrated by LMs across various tasks raise a wide concern about what they have learned in the representation space. An important and effective approach for interpreting and analyzing representations of LMs is linear probing [67, 87]. The main idea of linear probing is simple: training linear classifiers to predict some specific attributes or concepts (*e.g.,* lexical structure [9] ) from the representations in the hidden layers of LMs. A high probing result (*e.g.,* classification accuracy on the out-of-sample test set) tends to imply relevant information has been implicitly encoded in the representation space of LMs, although this does not imply LMs directly use these representations [67, 10]. Recent studies empirically demonstrate that concepts such as color [88], game states [8]. and geographic position are encoded in LMs. Furthermore, these concepts may even be linearly encoded in the representation space of LMs [8, 89].

**Collaborative filtering.** Collaborative filtering (CF) [90] is an advanced technique in modern recommender systems. The prevailing CF methods tend to adopt an ID-based paradigm, where users and items are typically represented as one-hot vectors, with an embedding table used for lookup [54]. Usually, these embedding parameters are learned by optimizing specific loss functions to reconstruct the history interaction pattern [68]. Recent advances in CF mainly benefit from two aspects, graph convolution [72] and contrastive learning [90]. These CF models exhibit superior recommendation performance by conducting the embedding propagation [52, 55] and applying contrastive learning objectives [80, 62, 66]. However, although effective, these methods are still limited, due to the ID-based paradigm. Since one-hot vectors contain no feature information beyond being identifiers, it is challenging to transfer pre-trained ID embeddings to other domains [37] or to leverage leading techniques from computer vision (CV) and natural language processing (NLP) [34].

**LMs for recommendation.** The remarkable language understanding and reasoning ability shown by LMs has attracted extensive attention in the field of recommendation. The application of LMs in recommendation can be categorized into three main approaches: LM-enhanced recommendation, LM as the modality encoder, and LLM-based recommendation. The first research direction, LLM-enhanced recommendation, focuses on empowering traditional recommenders with the semantic representations from LMs [48, 47, 46, 49, 91, 92]. Specifically, these methods introduce representations from LMs as additional features for traditional ID-based recommenders, to capture complicated user preferences. The second research line lies in adopting the LM as the text modality encoder, which is also known as a kind of modality-based recommendation (MoRec) [34, 35]. These methods tend to train the LM as the text modality encoder together with the traditional recommender. In previous studies, BERT-style LMs are widely used as the text modality encoder. The third research line, LLM-based recommendation, directly uses LLMs as the recommender and recommends items in a text generation paradigm. Early attempts focus on adopting in-context learning (ICL) [93] and prompting pre-trained LLMs [94–97]. However, such naive methods tend to yield poor performance compared to traditional models. Therefore, recent studies concentrate on fine-tuning LLMs on recommendation-related corpus [16, 15, 26, 25, 29] and align the LLMs with the representations from traditional recommenders as the additional modality [17, 20, 27, 98].

# B Linear Mapping

## B.1 Brief of Used LMs

We briefly introduce the LMs we use for linear mapping in Section 2.

- **BERT** [4] is an encoder-only language model based on the transformer architecture [3], pre-trained on text corpus with unsupervised tasks. BERT adopts bidirectional self-attention heads to learn bidirectional representations.

- **RoBERTa** [5] is an enhanced version of BERT. RoBERTa preserves the architecture of BERT but improves it by training with more data and large batches, adopting dynamic masking, and removing the next sentence prediction objective.

- **Llama2-7B** [6] is an open-source decoder-only LLM with 7 billion parameters. Llama2 adopts grouped-query attention, with longer context length and larger size of the pre-training corpus compared with Llama-7B [99].

Table 6: Linear mapping performance of randomly shuffled item representations

| | Books | | | Movies & TV | | | Video Games | | |
|---|---|---|---|---|---|---|---|---|---|
| | Recall | NDCG | HR | Recall | NDCG | HR | Recall | NDCG | HR |
| **BERT** | 0.0226 | 0.0194 | 0.1240 | 0.0415 | 0.0399 | 0.2362 | 0.0524 | 0.0309 | 0.1245 |
| **text-embeddings-3-large (Random)** | 0.0200 | 0.0197 | 0.1316 | 0.0559 | 0.0528 | 0.3204 | 0.0562 | 0.0328 | 0.1351 |
| **text-embeddings-3-large** | 0.0735 | 0.0608 | 0.3355 | 0.1109 | 0.1023 | 0.5200 | 0.1367 | 0.0793 | 0.2928 |

Table 7: Dataset statistics.

| | Books | Movies & TV | Video Games | Industrial | MovieLens-1M | Book Crossing |
|---|---|---|---|---|---|---|
| #Users | 7,176 | 14,382 | 40,834 | 15,141 | 6,040 | 6,273 |
| #Items | 10,728 | 1,000 | 14,344 | 5,163 | 3,043 | 5,335 |
| #Interactions | 1,304,453 | 129,748 | 390,013 | 82,578 | 995,492 | 253,057 |
| Density | 0.0169 | 0.0090 | 0.0701 | 0.0010 | 0.0542 | 0.0076 |

- **Mistral-7B** [69] is an open-source pre-trained decoder-only LLM with 7 billion parameters. Mistral 7B leverages grouped-query attention, coupled with sliding window attention for faster and lower cost inference.

- **text-embedding-ada-v2 & text-embeddings-3-large** [70] are leading text embedding models released by OpenAI. These models are built upon decoder-only GPT models, pre-trained on unsupervised data at scale with contrastive learning objectives.

- **SFR-Embedding-Mistral** [71] is a decoder-based text embedding model built upon the open-source LLM Mixtral-7B [69]. SFR-Embedding-Mistral introduces task-homogeneous batching and computes contrastive loss on "hard negatives", which brings a better performance than the vanilla Mixtral-7B model.

## B.2  Extracting Representations from LMs

We present how to extract representations from LMs. For encoder-based LMs (*e.g.,* BERT [4] and RoBERTa [5]), we use the output representation corresponding to the [CLS] token [40]. For decoder-based models (*e.g.,* Llama-7B [6, 69], Mistral-7B, and SFR-Embedding-Mistral [71]), we use the representation in the last transformer block [3], corresponding to the last input token [10, 100, 70]. Especially, for the commercial closed-source model (*e.g.,* text-embedding-ada-v2 and text-embeddings-3-large [2] [70]), we directly call the API interface to obtain representations.

## B.3  Empirical Findings

We find more evidence about representations in leading LM encode collaborative signals beyond better feature encoding ability. We randomly shuffle item representations and conduct the same linear mapping experiment. As illustrated in Table 6, randomly shuffled representations, text-embeddings-3-large (Random), yield similar performance with BERT, lagging largely behind the vanilla linear mapping method. These results indicate that BERT may only serve as a good feature encoder, while the latest LM may further encode collaborative signals beyond naive feature encoding.

# C  Experiments

## C.1  Datasets

We incorporate six datasets in this paper, including four datasets from the Amazon platform [3] [1] (*i.e.,* Books, Movies & TV, Video Games, and Industrial), and two datasets from other platforms (*i.e.,* MovieLens-1M and Book Crossing). Table 7 reports the data statistics of each dataset.

We divide the history interaction of each user into training, validation, and testing sets with a ratio of 4:3:3, and remove users with less than 20 interactions following previous studies [50]. We also remove items from the testing and validation sets that do not appear in the training set, to address the cold start problem.

---

[2] https://platform.openai.com/docs/guides/embeddings
[3] www.amazon.com

> **Item Title Examples**
>
> **Books:** *Dismissed with Prejudice: A J.P. Beaumont Novel*; *Die for Love: A Jacqueline Kirby Novel of Suspense*; *The Cloud*; *Memories Before and After the Sound of Music: An Autobiography*; *Harry Potter and the Sorcerer's Stone;*
> **Movies & TV:** *Batman Begins*; *Fantastic Four*; *Max Headroom: The Complete Series*; *Madagascar*; *Land of the Dead*; *King Kong;*
> **Video Games:** *Fighting Force*; *Tomb Raider II*; *Tomb Raider*; *WWF Warzone*; *Kartia: The Word of Fate*; *Snowboard Kids*; *Command & amp; Conquer: Tiberian Sun - PC*; *Final Fantasy VII*; *Grim Fandango - PC*; *Half-Life - PC;*
> **MovieLens-1M:** *Basquiat (1996)*; *Tin Cup (1996)*; *Godfather, The (1972)*; *Supercop (1992)*; *Manny & Lo (1996)*; *Bound (1996)*; *Carpool (1996);*
> **Book Crossing:** *Prague : A Novel*; *Chocolate Jesus*; *Wie Barney es sieht*; *To Kill a Mockingbird*; *Sturmzeit. Roman*; *A Soldier of the Great War*; *Pride and Prejudice (Dover Thrift Editions);*
> **Industrial:** *Jurassic Perisphinctes Ammonites from France*; *FS9140: Spinosaurus - Dinosaur Tooth 20-30mm*; *FS9410: USA Eocene, Fossil Fish (Knightia alt), A-grade*; *Delta 50-857 Charcoal Filter for 50-868*; *Hitachi RP30SA 7-1/2 Gallon Stainless Steel Industrial Shop Vacuum (Discontinued by Manufacturer)*; *Makita 632002-4 14-Inch Cut-Off Wheels (5-Pack) (Discontinued by Manufacturer)*; *PORTER-CABLE 740001801 4 1/2-Inch by 10yd 180 Grit Adhesive-Backed Sanding Roll;*

Figure 4: Example of item titles.

In this paper, we only use the item titles as the text description. Figure 4 gives some item title examples from different datasets.

## C.2 General Recommendation

### C.2.1 Baselines

We incorporate a series of CF models as our baselines for general recommendation. These models are classified as classical CF methods (MF, MultVAE, and LightGCN), CL-based CF methods (SGL, BC Loss, and XSimGCL), and LM-enhanced CF methods (KAR, RLMRec). For these LM-enhanced CF methods, we adopt the leading CF method XSimGCL as the backbone.

- **MF** [54, 68] is the most basic CF model. It denotes users and items with ID-based embeddings and conducts matrix factorization with Bayesian personalized ranking (BPR) loss.

- **MultVAE** [63] is a traditional CF model based on the variational autoencoder (VAE). It regards the item recommendation as a generative process from a multinomial distribution and uses variational inference to estimate parameters. We adopt the same model structure as suggested in the paper: $600 \rightarrow 200 \rightarrow 600$.

- **LightGCN** [55] is a light graph convolution network tailored for the recommendation, which deletes redundant feature transformation and activation function in NGCF [52].

- **SGL** [80] introduces graph contrastive learning into recommender models for the first time. By employing node or edge dropout to generate augmented graph views and conduct contrastive learning between two views, SGL achieves better performance than LightGCN.

- **BC Loss** [76] introduces a robust and model-agnostic contrastive loss, handling various data biases in recommendation, especially for popularity bias.

- **XSimGCL** [66] directly generates augmented views by adding noise into the inner layer of LightGCN without graph augmentation. The simplicity of XSimGCL leads to a faster convergence speed and better performance.

- **KAR** [48] enhances recommender models by integrating knowledge from large language models (LLMs). It generates textual descriptions of users and items and combine the LM representations with traditional recommenders using a hybrid-expert adaptor.

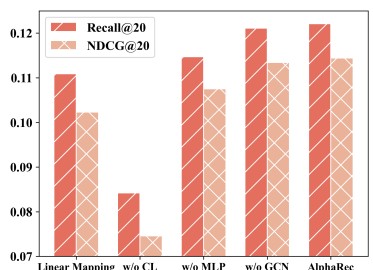

(a) Ablation study on Movies & TV

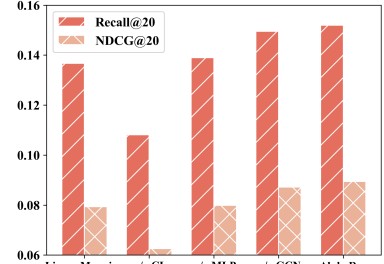

(b) Ablation study on Video Games

Figure 5: Ablation study

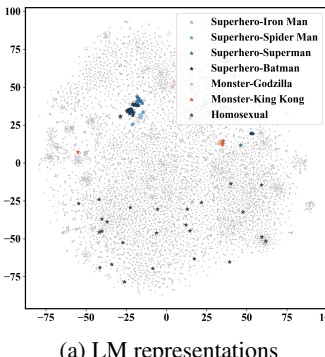

(a) LM representations

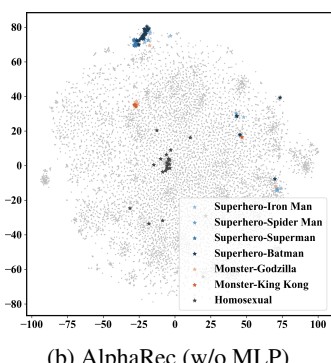

(b) AlphaRec (w/o MLP)

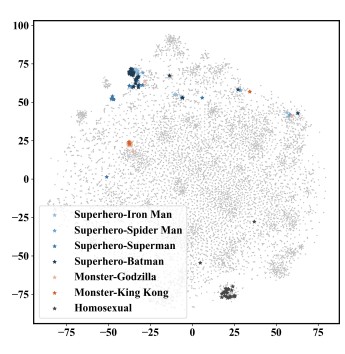

(c) AlphaRec

Figure 6: The t-SNE visualization of representations on Movies & TV. (6a) The item representations in the LM space. (6b) The item representations obtained by replacing the MLP with a linear mapping matrix in AlphaRec. (6c) The item representations obtained from AlphaRec.

- **RLMRec** [47] aligns semantic representations of users and items with the representations in CF models through a contrastive loss, as an additional loss trained together with the CF model. The fusion of semantic information and collaborative information brings performance improvement.

### C.2.2 Ablation Study

We conduct the same ablation study as introduced in Section 4.1 on Movies & TV and Video Games datasets. As illustrated in Figure 5, each component in AlphaRec contributes positively, which is consistent with our findings in Section 4.1.

### C.2.3 The t-SNE Visualization Comparison

In this section, we aim to intuitively explore how the MLP in AlphaRec further helps in excavating collaborative signals in language representations, compared to the linear mapping matrix. We visualize the item representations from LMs, AlphaRec (w/o MLP), and AlphaRec in Figure 6, where AlphaRec (w/o MLP) denotes replacing the MLP with a linear mapping matrix. We observed that movies about superhero and monster cluster in all representation spaces, indicating both AlphaRec (w/o MLP) and AlphaRec capture the preference similarities between these items and preserve the clustering relationship. The difference between AlphaRec (w/o MLP) and AlphaRec may lie in the ability to capture obscure preference similarities among items. As shown in Figure 6a, homosexual movies are dispersed in the language space, indicating the possible semantic differences between them. AlphaRec successfully captures the preference similarities and gathers these items in the representation space, while AlphaRec (w/o MLP) remains some items dispersed. Moreover, AlphaRec outperforms AlphaRec (w/o MLP) by a large margin, as indicated in Figure 5a. These results indicate that AlphaRec exhibits a more fine-grained preference capture ability with the help of nonlinear transformation.

Table 8: The effect of the training dataset on zero-shot recommendation

| | Industrial | | | MovieLens-1M | | | Book Crossing | | |
|---|---|---|---|---|---|---|---|---|---|
| | Recall | NDCG | HR | Recall | NDCG | HR | Recall | NDCG | HR |
| **AlphaRec (trained on Books)** | 0.0896 | 0.0562 | 0.1256 | 0.1218 | 0.2619 | 0.8942 | 0.0646 | 0.0532 | 0.3346 |
| **AlphaRec (trained on Movies & TV)** | 0.0909 | **0.0581** | 0.1266 | 0.1438 | 0.3122 | 0.9200 | 0.0471 | 0.0406 | 0.2600 |
| **AlphaRec (trained on Video Games)** | 0.0905 | 0.0567 | 0.1225 | 0.1221 | 0.2313 | 0.9034 | 0.0412 | 0.0378 | 0.2585 |
| **AlphaRec (trained on mixed dataset)** | **0.0913** | 0.0573 | **0.1277** | **0.1486** | **0.3215** | **0.9296** | **0.0660** | **0.0545** | **0.3381** |

Table 9: Performance comparison between training on the single dataset and the mixed dataset

| | Books | | | Movies & TV | | | Video Games | | |
|---|---|---|---|---|---|---|---|---|---|
| | Recall | NDCG | HR | Recall | NDCG | HR | Recall | NDCG | HR |
| **AlphaRec (trained on single dataset)** | **0.0991** | **0.0828** | **0.4185** | **0.1221** | **0.1144** | **0.5587** | **0.1519** | **0.0894** | **0.3207** |
| **AlphaRec (trained on mixed dataset)** | 0.0979 | 0.0818 | 0.4147 | 0.1194 | 0.1107 | 0.5463 | 0.1381 | 0.0827 | 0.2985 |

## C.3 Zero-shot Recommendation

### C.3.1 Co-training on Multiple Datasets

Co-training on multiple datasets is similar to training on one single dataset, where the only difference lies in the negative sampling. When co-training on multiple datasets, the negative items are restricted to the same dataset as the positive item rather than the full item pool. The other training procedures remain the same with training on one single dataset.

### C.3.2 Baselines

Since previous works about zero-shot recommendation mostly focus on sequential recommendation [83, 82], we slightly modify two methods in sequential recommendation, ZESRec [38] and UniSRec [37] as our baselines. Specifically, we maintain the model structure as provided in the paper, and adopt the training paradigm of CF.

- **Random** denotes randomly recommending items from the entire item pool.

- **Pop** denotes randomly recommending from the most popular items. Here popularity denotes the number of users that have interacted with the item.

- **ZESRec** [38] is the first work that defines the problem of zero-shot recommendation. To address this problem, this work introduces a hierarchical Bayesian model with representations from the pre-trained BERT.

- **UniSRec** [37] aims to learn universal item representations from BERT, with parametric whitening and a MoE-enhanced adaptor. By pre-training on multiple source datasets, UniSRec can conduct zero-shot recommendation on various datasets in a transductive or inductive paradigm.

### C.3.3 The Effect of Training Datasets

**The effect of the training dataset on zero-shot recommendation.** We report the zero-shot recommendation performance differences trained on different datasets in Table 8. Here AlphaRec (trained on Books) denotes training on a single Books dataset, while AlphaRec (trained on mixed dataset) denotes co-training on three Amazon datasets. Generally, training on more datasets lead to a better zero-shot performance.

**The performance comparison between training on the single dataset and the mixed dataset.** In Table 9, AlphaRec (trained on single dataset) denotes training and testing on the same single dataset, while AlphaRec (trained on mixed dataset) denotes training on three Amazon datasets and testing on one single dataset. Generally, co-training on three Amazon datasets yields similar performance compared with training on one single dataset. The only exception lies in Video Games, which shows some performance degradation. We attribute this to the difference between the selection of $\tau$. We use $\tau = 0.15$ when trained on the mixed dataset, while the optimal $\tau$ for Video Games lies around 0.2. These results indicate that a single AlphaRec can capture user preferences among various datasets, showcasing a general collaborative signal capture ability.

 ## C.4 User Intention Capture

 ### C.4.1 Intention Query Generation

---

**Intention Query Generation**

**Input**
You are an expert in generating queries for a target movie. Please help me generate the most suitable query for the target movie within one sentence, following the given example.
Example:
TARGET: BUG-A-SALT 3.0 Black Fly Edition.
QUERY: I want a gun that I can use while gardening to get rid of stink bugs, ants, flies, and spiders in my house. It needs to be amazing and help me feel less scared.
TARGET: Toy Story (1995).

**Output**
QUERY: I'm looking for a heartwarming animated movie that follows the adventures of a group of toys who come to life when their owner is not around.

---

Figure 7: Example of item query generation.

The user intention query is a natural language sentence implying the target item of interest. For each item in the dataset, we generate a fixed user intention query. Following the previous work [40], we generate user intention queries with the help of ChatGPT [85]. As shown in Figure 7, we prompt ChatGPT in a Chain-of-Thought (CoT) [101] paradigm and adopt the output as the user intention query. We adopt a rule-based strategy to ensure that the output query is in first person, and regenerate the wrong query. Considering the huge amount of item title text, we use ChatGPT3.5 API for generating all queries for the budget's sake.

### C.4.2 Baseline

AlphaRec exhibits user intention capture abilities, although not specially designed for search tasks. We compare AlphaRec with TEM [86] which falls in the field of personalized search [84, 102].

- **TEM** [86] uses a transformer to encode the intention query together with user history behaviors, which enables it to achieve better search results by considering the user's historical interest.

### C.4.3 Case Study

We conduct two more case studies to verify the user intention capture ability of AlphaRec. As illustrated in Figure 8 and Figure 9, AlphaRec provides proper recommendation results, including the target item for the user intention at the top.

### C.4.4 Effect of the Intention Strength Alpha

The value of $\alpha$ controls the balance between the user's historical interests and the user intention query. A larger $\alpha$ incorporates more about the user intention while considering less about the user's historical interests. As shown in Figure 10, the effect of $\alpha$ on Video Games shows a similar trend with MovieLens-1M.

## C.5 Trainig Cost

We report the training cost of AlphaRec in this section. Table 10 reports the seconds needed per epoch and the total training cost until convergence. Here Amazon-Mix denotes the mixed dataset of Books, Movies & TV, and Video Games. It's worth noting that AlphaRec converges quickly and only requires a small amount of training time.

**User Intention:** I'm looking for a quirky superhero movie with a comedic twist that features a group of misfit heroes with unusual powers trying to save the day.
**Target:** Mystery Men (1999)

**Recommendation List (w/o Intention)**
- American Pie (1999)
- Bug's Life, A (1998)
- General's Daughter, The (1999)
- Double Jeopardy (1999)
- Election (1999)

**Recommendation List (w Intention)**
- Mystery Men (1999)
- American Pie (1999)
- Bug's Life, A (1998)
- Election (1999)
- Wild Wild West (1999)

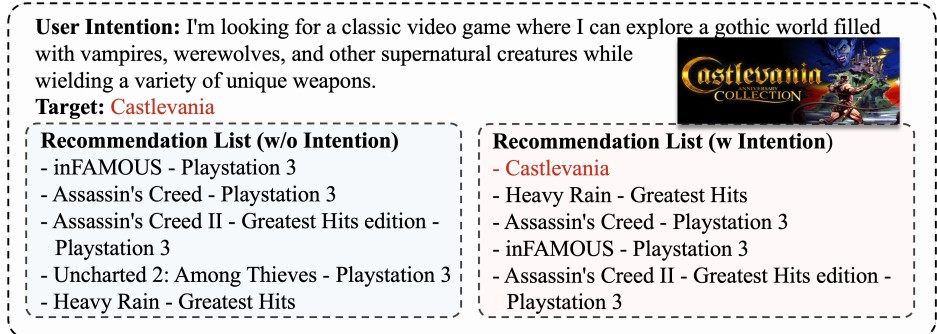

Figure 8: Case study of user intention capture on MovieLens-1M

**User Intention:** I'm looking for a classic video game where I can explore a gothic world filled with vampires, werewolves, and other supernatural creatures while wielding a variety of unique weapons.
**Target:** Castlevania

**Recommendation List (w/o Intention)**
- inFAMOUS - Playstation 3
- Assassin's Creed - Playstation 3
- Assassin's Creed II - Greatest Hits edition - Playstation 3
- Uncharted 2: Among Thieves - Playstation 3
- Heavy Rain - Greatest Hits

**Recommendation List (w Intention)**
- Castlevania
- Heavy Rain - Greatest Hits
- Assassin's Creed - Playstation 3
- inFAMOUS - Playstation 3
- Assassin's Creed II - Greatest Hits edition - Playstation 3

Figure 9: Case study of user intention capture on Video Games

# D Hyperparameter Settings and Implementation Details

We conduct all the experiments in PyTorch with a single NVIDIA RTX A5000 (24G) GPU and a 64 AMD EPYC 7543 32-Core Processor CPU. We optimize all methods with the Adam optimizer. For all ID-based CF methods, we set the layer numbers of graph propagation by default at 2, with the embedding size as 64 and the size of sampled negative items $|\mathcal{S}_u|$ as 256. We use the early stop strategy to avoid overfitting. We stop the training process if the Recall@20 metric on the validation set does not increase for 20 successive evaluations. In AlphaRec, the dimensions of the input and output in the two-layer MLP are 3072 and 64 respectively, with the hidden layer dimension as 1536. We apply the all-ranking strategy [103] for all experiments, which ranks all items except positive ones in the training set for each user. We search hyperparameters for baselines according to the suggestion in the literature. The hyperparameter search space is reported in Table 11. For these LM-enhanced models, KAR and RLMRec, we also search the hyperparameter of their backbone XSimGCL.

For AlphaRec, the only hyperparameter is the temperature $\tau$ and we search it in [0.05, 2]. We report the temperature $\tau$ we used for each dataset in Table 12. For the mixed dataset Amazon-Mix in Section 4.2, we use a universal $\tau = 0.15$. We adopt $\tau = 0.2$ for the MovieLens-1M dataset for the user intention capture experiment in Section 4.3.

# E Broader Impact

The proposed AlphaRec can significantly improve the performance of zero-shot recommendation and the capability of user intent capture, offering a good approach to crafting more personalized recommendation results. One concern of AlphaRec is the potential for the representations generated by language models can be maliciously attacked, which may result in erroneous or unexpected recommendations. Therefore, we kindly advise researchers to cautiously check the quality of the language representations before using AlphaRec.

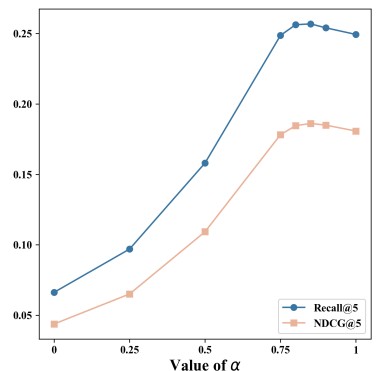

Figure 10: Effect of $\alpha$ on Video Games

Table 10: Training cost of AlphaRec (seconds per epoch/in total).

|          | Books        | Movies & TV | Video Games | Amazon-Mix     |
|----------|--------------|-------------|-------------|----------------|
| AlphaRec | 40.1 / 1363.4 | 12.3 / 479.7 | 7.4 / 214.6 | 107.2 / 5788.8 |

Table 11: Hyperparameters search spaces for baselines.

|              | Hyperparameter space |
|--------------|----------------------|
| **MF** & **LightGCN** | lr $\sim$ {1e-5, 3e-5, 5e-5, 1e-4, 3e-4, 5e-4, 1e-3} |
| **MultVAE** | dropout ratio $\sim$ {0, 0.2, 0.5}, $\beta \sim$ {0.2, 0.4, 0.6, 0.8} |
| **SGL** | $\tau \sim$ [0.05, 2], $\lambda_1 \sim$ {0.005, 0.01, 0.05, 0.1, 0.5, 1.0}, $\rho \sim$ {0, 0.1, 0.2, 0.3, 0.4, 0.5} |
| **BC Loss** | $\tau_1 \sim$ [0.05, 3], $\tau_2 \sim$ [0.05, 2] |
| **XSimGCL** | $\tau \sim$ [0.05, 2], $\epsilon \sim$ {0.01, 0.05, 0.1, 0.2, 0.5, 1.0}, $\lambda \sim$ {0.005, 0.01, 0.05, 0.1, 0.5, 1.0}, $l* = 1$ |
| **KAR** | No. shared experts $\sim$ {3, 4, 5}, No. preference experts $\sim$ {4, 5} |
| **RLMRec** | kd weight $\sim$ [0.05, 2], kd temperature $\sim$ [0.01, 0.05, 0.1, 0.15, 0.2, 0.5, 1] |
| **ZESRec** | $\lambda_u \sim$ {0.01, 0.05, 0.1, 0.5, 1.0}, $\lambda_v \sim$ {0.01, 0.05, 0.1, 0.5, 1.0} |
| **UniSRec** | lr $\sim$ {3e-4, 1e-3, 3e-3, 1e-2} |
| **TEM** | $l \sim$ {2,3}, head $h \sim$ {4, 8} |
| **AlphaRec** | $\tau \sim$ [0.05, 2] |

Table 12: The hyperparameters of AlphaRec

|        | Books | Movies & TV | Video Games | Amazon-Mix |
|--------|-------|-------------|-------------|------------|
| $\tau$ | 0.15  | 0.15        | 0.2         | 0.15       |

