# OpenReview forum: "Language Models Encode Collaborative Signals in Recommendation"
_NeurIPS.cc/2024/Conference — Submitted to NeurIPS 2024_

### Official Review · Reviewer_EzpW · 2024-07-01

**Soundness:** 2
**Presentation:** 3
**Contribution:** 2
**Rating:** 4
**Confidence:** 4

**Summary:**

The paper studies an important and open question, how much user behavior knowledge (generally captured by collaborative filtering models) are present in large language models. This has been a topic attracting significant research interest in recent years. The authors propose that simple linear mappings done on top of LM encoder representations are sufficient to capture collaborative filtering signals in recommendations, and propose a new recommendation method, AlphaRec, which takes pretrained language model content embeddings as input, transforms them via MLPs and lightweight graph convolutions, followed by a contrastive loss. The authors conduct experimental analysis for AlphaRec in both standard settings and zero-shot settings.

**Strengths:**

**S1**: the topic studied is important. It is generally believed that language model and collaborative filtering (recommendations) learn different representation spaces, and methods to bring the two spaces closer is of significant interest to the large community working on search, recommendations, ads, and related topics.

**S2**: the particular approach proposed (linear mapping from textual space to collaborative filtering space) is understudied in prior work on LLM and (Generative) CF, despite numerous papers in recent years.

**Weaknesses:**

**W1**: the writing in this paper, esp. recommendation system paradigm related discussions, misrepresents (or ignores) significant prior work done in the field. e.g., "AlphaRec follows a new CF paradigm, which we term the language-representation-based paradigm." and related writings.

- Content signals and/or embeddings have been used as the dominant recommendation paradigm in the field, even well before the seminal YouTube DNN paper [1] was published (see e.g., Pazzani and Billsus, 2007 [12]). For recent examples of related work, see e.g., [10, 11] from Pinterest and Meta in KDD'22 (but one should be able to easily find similar papers in WWW KDD etc in prior years as well).
- Replacing ResNet/ViT- or GPT-/BERT- generated embedding with LLaMa- or Mixtral- generated embeddings cannot and should not be viewed as a paradigm shift, especially given the core architecture of AlphaRec is not substationally different from prior work.

**W2**: AlphaRec needs to be compared with stronger baselines. This applies to many major experiments in the paper. Here are some examples of baselines missing, which may significantly change conclusions obtained and discussions etc:

- vs ID-based recommenders (Table 3).
    - Equation (2) and line 171-173 for $N_u$ already captures set of items that a user is related to ("user interaction history" / "user engagement history") to a large extent. Thus, the authors should compare AlphaRec with at least some SotA sequential/generative recommenders, such as SASRec, BERT4Rec, TIGER, HSTU [3, 4, 6, 7]. All of them are missing in the current paper.
    - Given AlphaRec uses the transposed item id representation - the one layer $N_i$ formulation (equation (2)), relevant work in recent years include Dual contrastive network [8] and User-centric ranking [9]. The authors should compare with or at least discuss some work in this category as related work.
- Zero-shot performance. (Table 4)
    - "Book Crossing" is not a commonly used benchmark dataset. The "Industrial" dataset (per citation [1] on line 273) seems to be a small-scale "Yelp" dataset, and should be renamed to avoid confusions.
    - For ML-1M, the SotA approach one year ago (LLMRank [14]) already achieved 53.73 NDCG@20, significantly higher compared with 32.15  (AlphaRec) in this work.

**W3**: many other formulations/experiments/writings could be significantly improved. Examples include:

- The proposed task formulation does not reflect how recommendation systems work in practice. e.g., "Line 97-99. Personalized item recommendation with implicit feedback aims to select items i ∈ I that best match user u’s preferences based on binary interaction data Y = [yui], where yui = 1 (yui = 0) indicates user u ∈ U has (has not) interacted with item i [58]." -- here "selecting the item that user will interact with" is not the same as "selecting the item with the highest reward", as the interaction itself can be negative (e.g., disliking a recommendation, abandoning session, etc.). See [1, 2] for references.

- A key contribution of this work should be the linear mapping finding. But Table 1 uses a questionable set of baselines for both LMs and CF baselines, which weakens linear mapping related claims.
    - To claim "Moreover, with the advances in LMs, the performance of item representation linearly mapped from LMs exhibits a rising trend, gradually surpassing traditional ID-based CF models" -- I would expect the authors to compare with a single set of models (e.g., LLaMa-2 7B 13B 70B or GPT-3 1.3b 2.7b 6.7b 13b 175b) trained on identical data. As it stands, all models are trained and/or finetuned with different data, so a simpler hypothesis explaining the LM (Linear Mapping) trend is that people are including more and more data into LLM pretraining/finetuning stages, which happen to capture more and more aspects relevant to recommendations.
    - On the CF side, "MF" "MultiVAE" and "LightGCN" do not represent SotA baselines on Amazon Review datasets (see W2).

- Table 1. Please highlight the particular K used for Recall and HR metrics (hard to find in the paper, applies to other tables too).   Most work on recommendation models also report HR/NDCG/etc. over at least 2-3 Ks to help readers understand how metrics vary with different approaches.


- Table 4. [1] should not be labeled as an "Industrial" dataset. The cited paper (per line 273) is Ni et al. "Justifying Recommendations using Distantly-Labeled Reviews and Fine-Grained Aspects" which in turn seems to be an publicly available review dataset provided by Yelp. Please use appropriate language as the current writing leads readers to think that AlphaRec is an industrially deployed system. Please refer to industrial papers (e.g., KDD ADS track papers [2, 9, 11, 12]) for how to describe testings done on publicly available industrial sampled datasets (like Yelp), vs real deployments.

- Misc: Contrastive loss is widely used and should not be viewed as a contribution of AlphaRec. See [15, 11] etc.

**Questions:**

**Q1**: Given the paper makes very significant claims, I would strongly recommend setting up experiments that  properly corroborate those claims to avoid the paper appearing like overselling. Examples include:
* Given linear mapping is a very weak baseline for utilizing LLMs, comparison with recent LLM4Rec approaches including in-context learning, transfer learning, etc. like [14] is necessary.
* To justify LLMs can increasingly capture collaborating filtering signals, please fix a LLM baseline (eg LLaMa-2 or GPT-3 or Gemma) and vary parameter count (eg up to 70/175b) to show that collaborating filtering knowledge is captured with increased model complexity. Right now a simpler interpretation of Table 1 is that different models use more data, with more data used for recent LLM training.
* Please compare with appropriate CF baselines, incl popular sequential recommenders in recent years as discussed in W2.

These experiments would make the findings/conclusions significantly more convincing, and I'm very happy to change rating if linear mapping/AlphaRec remains a competitive approach after these experiments.



**Q2**: Please consider ways to reframe the CF paradigm discussions as it's unclear how AlphaRec differs from building GCNs or RNNs/Transformers on top of ResNet-/BERT-/GPT-based embeddings, which has been a popular baseline for a long time eg [5, 11, 12].


**References**:
- [1] Convington et al.  Deep Neural Networks for YouTube Recommendations. RecSys'16.
- [2] Zhou et al. Deep Interest Network for Click-Through Rate Prediction. KDD'18.
- [3] Kang et al. Self-Attentive Sequential Recommendation. ICDM'18.
- [4] Sun et al. BERT4Rec: Sequential Recommendation with Bidirectional Encoder Representations from Transformer. CIKM'19.
- [5] Hidasi et al. Session-based Recommendations with Recurrent Neural Networks. ICLR'16.
- [6] Rajput et al. Recommender Systems with Generative Retrieval. NeurIPS'23.
- [7] Zhai et al. Actions Speak Louder than Words: Trillion-Parameter Sequential Transducers for Generative Recommendations. ICML'24.
- [8] Lin et al. Dual contrastive network for sequential recommendation. SIGIR'22.
- [9] Zhao et al. Breaking the Curse of Quality Saturation with User-Centric Ranking. KDD'23.
- [10] Pancha et al. PinnerFormer: Sequence Modeling for User Representation at Pinterest. KDD'22.
- [11] Rangadurai et al. NxtPost: User To Post Recommendations In Facebook Groups. KDD'22.
- [12] Pazzani and Billsus.  Content-based recommendation systems. In The adaptive web. 2007.
- [13] Naumov et al. Deep Learning Recommendation Model for Personalization and Recommendation Systems. 2019.
- [14] Hou et al. Large Language Models are Zero-Shot Rankers for Recommender Systems. ECIR'24.
- [15] Klenitsky et al. Turning Dross Into Gold Loss: is BERT4Rec really better than SASRec? RecSys'23.

---

> ### Author Rebuttal · Authors · 2024-08-07
>
> **Response to Reviewer $\color{orange}{\text{EzpW}}$**
>
> Thanks for your acknowledgment of the importance of the topic we studied. We believe you are an industrial expert with a deep understanding of sequential recommendation. We greatly appreciate your careful reading of our paper and your responsible comments. Some of your comments can significantly improve our paper. Below are our responses to your comments. **Due to the character limitation, results are presented in the attached PDF.**
>
> > **Comment 1:  The discussion of CF paradigm**
>
> Thanks for your concerns. We do not aim to distort nor omit the previous important works using content information and embeddings, as we include these previous studies as the early explorations of the language-representation-based paradigm in line 209-212 and compare AlphaRec with previous works like UniSRec. The primary goal of the CF paradigm discussion here is to summarize this research line, including the previous important works, into a unified paradigm (since there is no acknowledged definition of such paradigm; although content-based recommendation has been used previously, it is not accurate for this paradigm), and showcase the advantage of using the advanced LM representations. In hindsight, we agree using the word "new" is a little glib. We wish to apologize for this overstatement.
>
> Additionally, we still address the importance of the shift from BERT-style representations to LLM-based representations, for three key factors: more encoded user preference similarities, superior zero-shot recommendation ability, and user intention-aware ability.
>
> > **Comment 2:  Appropriate experiments to verify the claim**
>
> Thanks for your comments. The mentioned experiments will help make our claim more convincing. We would like to explain the setting differences of our paper with sequential recommendation first. It's true that sequential recommendation (or next item prediction) is important. However, general recommendation [1] (or direct recommendation [2]), which aims to discover the general preferences of users rather than predicting the next item, is still an innegligible issue even recently [3, 4]. They are deployed in domains like music recommendation [3]. The motivation for adopting general recommendation in our paper is reasonable: we do not wish to introduce any non-linearity for studying the representation space relationship, while the sequential relationship is usually captured by non-linear transformation such as attention [5] or RNN. As a result, the general recommendation is an appropriate task for our research goal.
>
> 1. **Appropriate experiments and stronger baselines**
>
>    Considering the general recommendation setting we adopted, the baselines in our paper are the latest and SOTA methods (XSimGCL [3] and RLMRec [4]). We also compare the linear mapping method with these baselines in **Table 13**.
>
>    We also consider your suggestion about comparing linear mapping with sequential models in the sequential setting. The results are presented in **Table 14**. AlphaRec is not compared due to the GCN module is not designed for the sequential recommendation, making it hard to adapt to sequential recommendation.
>
>    It's worth noting that linear mapping does not contain any model design, but still maintains comparable performance with specially designed sequential models SASRec. These results are inspiring and make our claim more convincing.
>
> 2. **Fix LLM type and vary parameter size.**
>
>    We fix the model of Llama2 family and change the parameter size. We report the performance in **Table 15** in the PDF. A clear performance increase is observed as the model size increases.
>
> 3. **Zero-shot performance**
>
>    In our zero-shot setting, no candidate set is adopted. While in LLMRank [6], a candidate set with the size of 20 is used, which leads to the reported high performance. We follow the same setting of LLMRank, equipping 19 random negative items for each positive item, to conduct the zero-shot experiments again. As indicated in **Table 16**, AlphaRec significantly outperforms LLMRank.
>
> > **Comment 3:  Improvements in the details**
>
> Once again, thank you for your meticulous reading of our article.
>
> 1. **The task formulation**
>
>    It is true that "selecting the item with the highest reward" is more practical in industry. However, "selecting items that the user has interacted with" is also a widely used setting for academic research, such as the famous SASRec [4].
>
> 2. **The particular K used and 2-3 Ks**
>
>    Thanks for your suggestion. We use K = 20 in this paper, and we will highlight this in the latest version of this paper. For more Ks, we have reported more Ks results in our rebuttal period and we will add more Ks results in the appendix of our latest version.
>
> 3. **The Industrial dataset**
>
>    Thanks for your suggestion. This dataset is the Amazon Industrial dataset on Amazon Review. We use this citation because it is the official citation provided by the author of the dataset Amazon Review [7]. We will rename it as Amazon-Industrial for better clarification.
>
> 4. **The contrastive loss**
>
>    We do not regard contrastive loss as the contribution of our paper, as we have clarified this in line 149-151 and line 335-337. The contribution of our paper is that we prove that only simple model design can arouse the potential of advanced language representations.
>
> [1] Neural collaborative filtering. 2017 WWW.
>
> [2] Recommendation as language processing (rlp): A unified pretrain, personalized prompt & predict paradigm (p5). 2022 RecSys.
>
> [3] Representation Learning with Large Language Models for Recommendation. 2024 WWW.
>
> [4] XSimGCL: Towards extremely simple graph contrastive learning for recommendation. 2023 TKDE.
>
> [5] Self-attentive sequential recommendation. 2018 ICDE.
>
> [6] Large Language Models are Zero-Shot Rankers for Recommender Systems. 2024 ECIR.
>
> [7] Justifying recommendations using distantly-labeled reviews and fine-grained aspect. 2019 EMNLP.

---

> > ### Comment · Reviewer_EzpW · 2024-08-12
> > **clarifying questions**
> >
> > Thanks authors for the detailed responses and for the newly added experiments. I have some further clarifying questions.
> >
> > * For Table 15, do we project LLaMa-2 7B/13B/70B into hidden spaces of the same dimensionality or different dimensionalities? How are the hyperparameters used in "linear mapping" chosen?
> > * Also as tJs7 asked, how does the linear mapping conclusion change if we provide additional information (e.g., encode both title and descriptions)?

---

> > > ### Author Response · Authors · 2024-08-12
> > > **Response for new questions**
> > >
> > > Thanks for your continued interest in our work.
> > >
> > > For the first question, we use the completely same experiment setting for different LM sizes, with the only difference in the input language representations. These language representations are projected into the same dimensionality of 64, which is consistent with the dimension for ID-based methods. Excepting the tau in the InfoNCE loss, the rest of the hyperparameters of linear mapping are also the same: batch size = 4096, learning rate = 0.0005, seed = 101, negative sampling number =256. For tau, 0.15, 0.15, and 0.20 are used for Books, Movies & TV, and Video Games respectively.
> > >
> > > We also appreciate the suggestion of providing additional information. The response to this question is on the way, for the necessary time to set up this experiment. Thanks again.

---

> > > ### Author Response · Authors · 2024-08-12
> > > **Linear mapping performance with additional information**
> > >
> > > The mentioned experiment is important and interesting. We conduct experiments on Llama2-7B, with different item information types provided. We have tried two types of additional information: with brand & price and with description. We give examples of the input text with different item information types.
> > >
> > > Examples:
> > >
> > > **Title only:** Marvel Super Heroes
> > >
> > > **Title + Brand & Price:** Marvel Super Heroes. Price: *$79.99*. Brand: Capcom
> > >
> > > **Title + Description:** Marvel Super Heroes. Description: This successor to Capcom's wildly popular X-Men allows you to go one-on-one at home with the Marvel Super Heroes. Your super heroes battle through rounds of competition facing off with Dr. Doom and ultimately Thanos. Take control of the amazing Infinity Gem system which grants super-powers such as healing extra attacking strength and infinite super attacks. Master a variety of fantastic attacks super moves and tons of multi-hit combos. For comic book fanatics and fighting game warriors alike Marvel Super Heroes is a must-play masterpiece!
> > >
> > > We adopt the same linear mapping setting as before, with the only difference in the input features. We report the performance comparison in **Table 1**.
> > >
> > > Table 1 Performance of linear mapping when different item information types
> > >
> > > |                       |           | Movies & TV |        |           |  Games  |        |
> > > | :-------------------: | :-------: | :---------: | :----: | :-------: | :-----: | :----: |
> > > |                       | Recall@20 |   NDCG@20   | HR@20  | Recall@20 | NDCG@20 | HR@20  |
> > > |  Title + Description  |  0.0863   |   0.0807    | 0.4426 |  0.0962   | 0.0557  | 0.2199 |
> > > | Title + Brand & Price |  0.0924   |   0.0866    | 0.4679 |  0.1253   | 0.0733  | 0.2735 |
> > > |      Title only       |  0.1027   |   0.0955    | 0.4952 |  0.1249   | 0.0729  | 0.2746 |
> > >
> > > As illustrated in **Table 1**, adding additional information does not bring performance improvements. Instead, a performance drop is observed. This finding is consistent with the findings in the previous work in the NLP community [1]. Moreover, adding descriptions leads to more performance drop than adding brand & price.
> > >
> > > There may be several reasons for the performance drop:
> > >
> > > 1. Metadata is missing frequently in the Amazon Review datasets. Around 1/2 of items miss at least one type of information among brand, price, and description. The flaw of the dataset also leads to inevitable noise in the representations.
> > > 2. Overwhelmingly long descriptions may serve as noise. The description is always long compared with titles. In most cases, item titles can be identified by the LM. However, when adding descriptions, the identifiable item becomes obscure. Worse still, as the example shows, the descriptions in Amazon Review tend to be a kind of marketing slogan, which leads to more noise.
> > > 3. Advanced language models can identify the target item correctly, with no need for additional information. LMs may understand the item title in their feature space [1] through a large amount of training corpus (i.e., items are encoded as a kind of world knowledge).
> > >
> > > As a consequence, the noise from the additional information (e.g., metadata missing and overwhelmingly long descriptions) makes it hard to study the representation space relationship. So we only encode item titles in this paper.
> > >
> > > The above results are consistent with our response to reviewer tjs7 that "If the language model encodes sufficient world knowledge, it should be able to uniquely identify items using simple item titles. Therefore, we have ignored other descriptions of items to avoid introducing unnecessary noise."
> > >
> > > [1] Language models represent space and time. 2024 ICLR.

---

> > > > ### Comment · Reviewer_EzpW · 2024-08-13
> > > >
> > > > Thanks authors for the clarification and for the updated experiments. To me, Table 15 and the newly added Table 1 have significantly improved the soundness of the core claims in the paper. My 2c is that, such discussions should have been given more space in the original paper per title "language models encode collaborative signals in recommendations", and discussions such as "language-representation-based paradigms" or  GCNs in AlphaRec somewhat diluted the contributions of this work.
> > > >
> > > > I will increase the score given discussions. My remaining concerns are still related to certain strong claims in this paper and whether they have been sufficiently justified by experiments, including "language-representation-based recommendation systems" as a new paradigm (please do consider rephrasing in future revisions), "outperforms leading ID-based CF models" (this seems to somewhat depend on how we split datasets per experiments), and how we explain the fact that LLMs, which are often robust to noises in their inputs, struggle to map additional information to the same linear space.

---

> > > > > ### Author Response · Authors · 2024-08-13
> > > > > **Clarifying questions about your comments**
> > > > >
> > > > > Thanks for your comments. I believe that you are a responsible reviewer. Your suggestions have helped us significantly improve our paper. I also have some clarifying questions about your suggestions.
> > > > >
> > > > > 1. Are current experiments strong enough to support our main claim? As you mentioned, "Table 15 and the newly added Table 1 have significantly improved the soundness of the core claims". However, you still give a score of 2 for the soundness. I am curious about the reason for this, since you have also mentioned “My remaining concerns are still related to certain strong claims in this paper and whether they have been sufficiently justified by experiments”. These two comments seem conflicting. Are current experiments still weak or does the low score come from other reasons? If still weak, I am also curious about the reason.
> > > > >
> > > > > 2. “How we explain the fact that LLMs, which are often robust to noises in their inputs, struggle to map additional information to the same linear space.” I am curious about problems in my explanations for this, for which I believe is reasonable.
> > > > >
> > > > > 3. Your overall comments about our work. I am curious about the overall score of 4 after our discussions. Do you think this paper is important and convincing but just needs reorganization about the paper? Or are current experiments still not convincing enough or the overall quality of the research question does not meet the requirements?
> > > > >
> > > > > Moreover, I am also curious about your ratings if this paper is reorganized as you have mentioned: “such discussions should have been given more space in the original paper”.
> > > > > Thanks so much for your time and valuable suggestions again.

---

### Official Review · Reviewer_GXUC · 2024-07-10

**Soundness:** 2
**Presentation:** 3
**Contribution:** 2
**Rating:** 4
**Confidence:** 3

**Summary:**

This paper states that LLM encodes collaborative signals that make it easy to connect language representation space with an effective recommendation space. Thus, it proposes an effective collaborative filtering model AlphaRec that takes as input only the transformed LLM representations of textual descriptions of items and is trained by InfoNCE loss and graph neural networks. The proposed method outperforms traditional ID-based models and other LM-enhanced methods.

**Strengths:**

1. This paper is well-written and easy to follow.
2. The paper conducts extensive experiments validating the effectiveness of the methods and proves the validity of the design through ablation study and anlysis.
3. The proposed method exhibits significantly good zero-shot recommendation performance

**Weaknesses:**

1. One of the most important motivations of the work is that the paper declares large language models encode collaborative signals which indicates the advantage of using representations of large language models for recommendations compared to id embeddings. However, how the preliminary experiments prove this point is insufficiently discussed in the paper. Advanced large language encodes more semantics of the textual descriptions and thus yields better performance. Why this alone doesn't fully explain the performance gain of LLMs should be more explicitly discussed in the main paper.
2. The novelty is limited. Using semantic embeddings of items has been widely used in recommendations. The novelty mostly lies in using the representations of large language models and the implementation details of how to make it effective when combined with traditional recommendation frameworks like non-linear transformations.
3. The paper states that language representation-based methods have low training costs. Still, if taking into account the costs of generating language representations, the computational cost is much higher than ID-based methods.

**Questions:**

It states that the improvements of using a more advanced LLM when using linear mapping of the representations do not merely come from the better feature encoding ability. However, the validating experiments in Appendix B3 is not convincing enough. Randomly shuffling the item representations eliminates the semantic relations between user-item interaction pairs and makes language models unable to leverage their semantic understanding ability, which explains the decline of the performance. Could you please elaborate more on the points you make in Appendix B3?

---

> ### Author Rebuttal · Authors · 2024-08-07
>
> **Response to Reviewer $\color{blue}{\text{GXUC}}$**
>
> > **Comment 1: More discussion about the collaborative signals encoded in LMs**
>
> Thanks for your concern and suggestion. We respond to this question as follows.
>
> First. We would like to restate the importance and correctness of the linear mapping method we used. It is acknowledged by the NLP community that a high performance of linear methods (i.e., linear mapping [1] and linear probing [2]) would suggest specific signals have been encoded in advanced LMs [3, 4], since the linear structure ensures the *homomorphism* [5] between two spaces. With such linear methods, lexical [6] and syntax [7] structures have been found inside LMs. More details about this research line can be found in previous works [2, 3] on top-tier NLP conferences. Therefore, a high performance of linear mapping would suggest that collaborative signals are encoded in LMs.
>
> Second. "More semantics of the textual descriptions" does not fully explain the success of linear mapping. Linear mapping captures the user preferences beyond textual similarities. As shown in Figure 1c, homosexual movies such as *My Beautiful Laundrette* and *Food of Love* scatter in the language space, but gather together in the recommendation space. These movies are semantically dissimilar but share similar user preference similarities, which is captured by the linear mapping. Therefore, we can not simply attribute the improvement to more semantics.
>
> Third. We add more experiments about linear mapping according to the suggestion of another reviewer (results are presented in the attached PDF). We also consider the sequential recommendation setting in Table 14 and the effect of LM size in Table 15. These results further indicates that user preference similarities are encoded in advanced LM, and the knowledge inside increases as the model size increases.
>
> > **Comment 2: The novelty is limited**
>
> Thanks for your concern about the novelty of this paper. The novelty of this paper does not lie in replacing previous BERT-style embeddings with advanced LM embeddings, or deliberately designing a new CF module. We address the two key novelties of this paper.
>
> 1. We are the first work to use linear mapping to reveal how much knowledge about recommendation is encoded in language models. Although linear mapping and linear probing are widely used and recognized methods for exploring the knowledge embedded in language models in the NLP field [2, 3], relevant exploration in recommendation is rare.
>
> 2. We prove that advanced language representations exhibit excellent properties for designing recommenders with multiple abilities. With only a simple model design, language-representation-based CF models can surpass traditional ID-based models on multiple tasks. Moreover, we summarize the advantages of using current advanced LM representations: low training cost, superior zero-shot performance, and user intention-aware ability.
>
> > **Comment 3:  The computational cost is high**
>
> Thanks for your concern. We would like to reply to your concern from three aspects.
> First. We would like to kindly point out that the computational cost you mentioned does not equal the training cost in our paper. We state in our paper that AlphaRec is of low training cost rather than computational cost.
> Second. It is important to highlight that language representations can be computed and documented in advance, and there is only a one-time computational cost.
> Third. The time cost is comparable with ID-based methods. The training cost is relatively low, compared with LM-based recommenders. And the total time cost is comparable with ID-based methods. For quantitative analysis, we represent the training time cost comparison as follows.
>
> **Table 1 Training cost comparison**
>
> ||Books|Movies & TV|Games|
> |:-:|:-:|:-:|:-:|
> |LM-based Methods|hours|hours|hours|
> |LightGCN|5235.1s|1328.2s|769.7s|
> |XSimGCL|761.1s|205.6s|124.6s|
> |AlphaRec|1363.4s|479.7s|214.6s|
>
> As indicated in Table 1, AlphaRec maintains competitive training costs with ID-based methods, which are much lower than LM-based methods.
>
> > **Question 1:  Elaborate more on the points in Appendix B3**
>
> Thanks for your concern. We assume that the improvement of advanced LMs comes from two possible ways, better feature encoding ability (such as more compact feature space or higher embedding dimension) and any other features or knowledge about recommendation. By shuffling the embeddings, we eliminate any other features or knowledge about recommendation. In this way, the performance only comes from the feature encoding ability. Therefore, the decline in performance proves that the improvement not only comes from better feature encoding ability, which is consistent with the claim in our paper.
>
> To summarize, we follow the following points to prove user preference similarities (i.e., collaborative signals) are encoded in advanced LMs:
>
> 1. The linear mapping performance may come from feature encoding ability or knowledge about recommendation (e.g., textual similarity or preference similarity). Random shuffle results are poor -> The performance largely comes from the knowledge about recommendation.
> 2. The linear mapping performance is excellent -> Previous works in NLP [2, 3, 4, 6, 7] suggest user preference similarities may be encoded in advanced LMs.
> 3. Textually dissimilar items gather in the recommendation space -> user preference similarities beyond textual similarities are encoded in advanced LMs.
>
>
> [1] Linearly mapping from image to text space. 2023. ICLR.
>
> [2] Understanding intermediate layers using linear classifier probes. 2017 ICLR.
>
> [3] Language models represent space and time. 2024 ICLR.
>
> [4] Emergent world representations: Exploring a sequence model trained on asynthetic task. 2023 ICLR
>
> [5] Linear algebra and geometry. 1969
>
> [6] Probing pretrained language models for lexical semantics. 2020 EMNLP.
>
> [7] A Structural Probe for Finding Syntax in Word Representations. 2019 NAACL.

---

### Official Review · Reviewer_tJs7 · 2024-07-16

**Soundness:** 3
**Presentation:** 3
**Contribution:** 3
**Rating:** 7
**Confidence:** 5

**Summary:**

The paper proposes AlphaRec, a novel method to incorporate both knowledge from pre-trained language models and collaborative signals. Authors firstly reveal the advantages brought from pre-trained embedding model, and then propose three modules within AlphaRec. An MLP layer to transform pre-trained embedding to item-representation. A graph convolution to aggregate neighbor’s information, and the InfoNCE loss to train introduced parameters within the MLP for each dataset. Overall, the novelty of this paper lies within exploration of NLP encoded embedding on RecSys. The graph convolution and InfoNCE loss are already widely used techniques.

**Strengths:**

1. A good exploration on new direction (language-representation-based) RecSys
2. Experiments are conducted from different angles for analyzing their model.

**Weaknesses:**

1. Insufficient baselines.
2. Uncleared model name definition.

**Questions:**

1. Why do authors only encode titles? There is more information within your used Amazon dataset including item descriptions.

2. I personally do not prefer un-informative model names such as the AlphaRec in this paper. Authors mentioned “This model is named AlphaRec for its originality and a series of good properties”. The reason seems so strange to me.

3. More advanced baselines are needed to compare such as the DirectAU [1]  and GraphAU [2]

[1] Wang, C., Yu, Y., Ma, W., Zhang, M., Chen, C., Liu, Y., & Ma, S. (2022, August). Towards representation alignment and uniformity in collaborative filtering. In Proceedings of the 28th ACM SIGKDD conference on knowledge discovery and data mining (pp. 1816-1825).
[2] Yang, L., Liu, Z., Wang, C., Yang, M., Liu, X., Ma, J., & Yu, P. S. (2023, October). Graph-based alignment and uniformity for recommendation. In Proceedings of the 32nd ACM International Conference on Information and Knowledge Management (pp. 4395-4399).

**Limitations:**

See Weakness.

---

> ### Author Rebuttal · Authors · 2024-08-07
>
> **Response to Reviewer $\color{green}{\text{tJs7}}$**
>
> We sincerely thank you for the positive feedback and valuable comments! To address your concerns, we present our responses as follows.
>
> > **Comment 1: Why only encode titles** There is more information within your used Amazon dataset including item descriptions.
>
> Thanks for your concern. An important contribution of this paper is the investigation of the how much recommendation knowledge is encoded in LMs through linear mapping [1, 2, 3]. If the language model encodes sufficient world knowledge, it should be able to uniquely identify items using simple item titles. Therefore, we have ignored other descriptions of items to avoid introducing unnecessary noise.
>
> > **Comment 2: Un-informative model names** There is more information within your used Amazon dataset including item descriptions.
>
> Thanks for your suggestion! The model name AlphaRec is also known for the adjustable hyperparameter \alpha for user intention capture, as introduced in Section 4.3. We believe that this is more reasonable for the model name we used.
>
> > **Comment 3: More advanced baselines**
>
> Thanks for your great suggestions! We fully agree that considering more CF methods will further verify the effectiveness of AlphaRec, especially with the research line on the uniformity and alignment in recommendation. We compare AlphaRec with DiretAU and GraphAU. As shown in this table, AlphaRec presents relatively high performance compared with these methods across various datasets.
>
>
> |    || Books ||| Movies & TV ||| Games ||
> | :------: | :--: | :---: | :--: | :--: | :---------: | :--: | :--: | :---: | :--: |
> |  | Recall | NDCG | HR | Recall | NDCG | HR | Recall | NDCG | HR |
> | DirectAU | 0.0889 | 0.0734 | 0.3914 | 0.1011 | 0.0992 | 0.4984 | 0.1038 | 0.0635 | 0.2312 |
> | GraphAU  | 0.0933 | 0.0747 | 0.4101 | 0.1135 | 0.1054 | 0.5163 | 0.1231 | 0.0718 | 0.2511 |
> | AlphaRec | **0.0991** | **0.0828** | **0.4185** | **0.1221** | **0.1144** | **0.5587** | **0.1519** | **0.0894** | **0.3207** |
>
> [1] Linearly mapping from image to text space. 2023. ICLR.
>
> [2] Understanding intermediate layers using linear classifier probes. 2017 ICLR.
>
> [3] Language models represent space and time. 2024 ICLR.

---

> > ### Comment · Reviewer_tJs7 · 2024-08-12
> > **Keep score unchanged**
> >
> > I will keep my score unchanged as 7 score is already qualified.

---

> > > ### Author Response · Authors · 2024-08-12
> > > **Thanks!**
> > >
> > > Thank you for your positive comments and valuable feedback! Your comments significantly help us improve our paper.

---

### Official Review · Reviewer_NfrM · 2024-07-18

**Soundness:** 2
**Presentation:** 2
**Contribution:** 2
**Rating:** 4
**Confidence:** 3

**Summary:**

This paper proposes AlphaRec, an LLM-based recommender system that utilizes language representations of item textual data for recommendations.

**Strengths:**

+ Investigating ID paradigm and LLM paradigm is important.
+ The method is simple but seems to be effective.

**Weaknesses:**

- In this paper, what most confuses me is the usage of the terminology "collaborative filtering" throughout the paper. In traditional recommender system, collaborative filtering information means the interactions among users and items. The authors find that using LM as feature extractors to get user/item embeddings from meta-data can achieve similar results as if CF is used for recommendation. However, this seems to be fundamentally different than LM has the "collaborative information", as for most online service platforms, the interaction data should be confident and open source LMs won't be able to train on that data. Therefore, the main claim in the paper seems questionable.

- It would be beneficial if we could have results on more diverse datasets.

**Questions:**

Please refer to my summary of weakness.

---

> ### Author Rebuttal · Authors · 2024-08-07
>
> **Response to Reviewer $\color{red}{\text{NfrM}}$**
>
> We sincerely thank you for your concerns about our paper.
>
> > **Comment 1: Terminology** The usage of the terminology "collaborative filtering"
>
> Thanks for your question. We would like to kindly point out that collaborative filtering is a common paradigm in recommendation [1], and we only use this terminology to describe the paradigm we adopt in this paper.
>
> Additionally, we do not use the terminology of collaborative filtering information in our paper, and there is no formal definition of collaborative filtering information in the literature. I guess you may mean collaborative signals [2] rather than collaborative filtering information. In our paper, we describe the collaborative signals as "*user preference similarities between items*", which is consistent with the definition in the famous paper NGCF [2].
>
> In this way, our claim equals whether user preference similarities are encoded in advanced LMs. In this paper, we use linear mapping to study this claim. The linear mapping or linear probing method is widely used to explore the encoded feature in LMs [3, 4]. It is acknowledged by the NLP community that a high performance with a linear layer would suggest that specific knowledge has been encoded in LMs [5] (since linear mapping ensures the homomorphism [6] between two spaces). As indicated in our paper, linear mapping yields high recommendation performance, which suggests that user preference similarities may be implicitly encoded in advanced LMs. More details of the linear mapping and linear probing method can be found in previous works that have been published in top-tier NLP conferences [4, 7, 8].
>
> According to the suggestions of other reviewers, we also conduct more experiments on sequential recommendation and model size effect of LMs in the attached PDF. These results can better clarify our claim.
>
> > **Comment 2: Training data for LMs** The interaction data is confident and open-source LMs won't be able to train on that data
>
> Thanks for your concern. We would like to answer this question from two aspects.
>
> First. All the datasets we adopted in this paper are public datasets, and everyone has access to these data. Open-source LMs have probably been trained on such user behavior data. It's important to highlight that most of the training data for open-source LMs are crawled from websites, such as the CommonCrawl data [9] used for training the Llama family [10]. Moreover, some LMs clearly state that Amazon Review data is used for training [11].
>
> Second. Features in LMs do not require LMs to be trained on a specific type of data. Evidence for this comes from previous works that find that the lexical [8] and syntax [12] structure is encoded in LMs. Lexical and syntax data rarely appear in the training corpus explicitly, but LMs still learn such features through training on a huge amount of structured natural language data. From this aspect, LMs are also able to learn the preference similarities of items through training on the user behavior data.
>
> > **Comment 3:  The main claim in the paper seems questionable.**
>
> Thanks for your concern. We answer this question in our response to comment 2. Moreover, we use experiment results to verify our claim like previous papers [4, 5, 6, 7, 8].
>
> > **Comment 4: More diverse datasets** It would be beneficial if we could have results on more diverse datasets.
>
> Thanks for your comments. However, we would like to kindly point out that six datasets from three platforms (i.e., Amazon, MovieLens, and BookCrossing) have been adopted in our paper.
>
> The scale of this dataset, as well as the scale of our experiments, is quite large in the field of recommendation systems. Moreover, most recommender system papers (including highly influential papers) typically conduct experiments on only 2-3 datasets [13, 15, 16] and a single data platform (e.g., Amazon) [14, 15].
>
> According to your suggestion and comments of reviewer $\color{orange}{\text{EzpW}}$, we add one more experiment on the steam dataset under the zero-shot recommendation setting and compare the performance with the LLM4Rec baseline LLMRank [16]. We follow the task setting of LLMRank, equipping 19 negative items for each positive item and conducting the selection task. As indicated in Table 1, AlphaRec significantly outperforms LLMRank on the Steam dataset.
>
> **Table 1 Zero-shot performance on Steam dataset**
>
> |||Steam|||
> |:-:|:-:|:-:|:-:|:-:|
> ||NDCG@1|NDCG@5|NDCG10|NDCG@20|
> |LLMRank|0.3112|0.4413|0.5255|0.5302|
> |AlphaRec|**0.4450**|**0.6131**| **0.6394**|**0.6714**|
> | Imp. %|42.99%|38.93%|21.67%|26.63%|
>
> [1] Neural collaborative filtering. 2017 WWW.
>
> [2] Neural graph collaborative filtering. 2019 SIGIR.
>
> [3] Linearly mapping from image to text space. 2023. ICLR.
>
> [4] Understanding intermediate layers using linear classifier probes. 2017 ICLR.
>
> [5] Language models represent space and time. 2024 ICLR.
>
> [6] Linear algebra and geometry. 1969
>
> [7] Emergent world representations: Exploring a sequence model trained on asynthetic task. 2023 ICLR
>
> [8] Probing pretrained language models for lexical semantics. 2020 EMNLP.
>
> [9] CCNet: Extracting high quality monolingual datasets from web crawl data. 2020 LREC.
>
> [10] LLaMA: Open and Efficient Foundation Language Models. 2023
>
> [11] SFR-Embedding-Mistral: Enhance Text Retrieval with Transfer Learning. 2024
>
> [12] A Structural Probe for Finding Syntax in Word Representations. 2019 NAACL.
>
> [13] Self-attentive sequential recommendation. 2018 ICDE.
>
> [14] Towards universal sequence representation learning for recommender systems. 2022 KDD.
>
> [15] Recommender Systems with Generative Retrieval. 2024 NeurIPS.
>
> [16] Large language models are zero-shot rankers for recommender systems. 2024 ECIR.

---

### Author Rebuttal · Authors · 2024-08-07

We sincerely appreciate the efforts of every reviewer to make this paper better. We are delighted to see the importance of the studied topic in this paper is acknowledged by most of the reviewers ($\color{red}{\text{NfrM}}$, $\color{green}{\text{tJs7}}$, and $\color{orange}{\text{EzpW}}$).

We appreciate all the reviewers for their valuable comments and suggestions, which help us significantly improve this paper. We summarize our response and the updates to the paper as follows:

- **More detailed discussion about the collaborative signals encoded in advanced LMs.** Most of the concerns from reviewers concentrate on whether linear mapping indicates that collaborative signals are encoded in LMs. As acknowledged in the community of NLP, the high performance of linear mapping and linear mapping reflects that LM may have implicitly encoded specific knowledge inside.

  We follow the following points to prove user preference similarities (i.e., collaborative signals) are encoded in advanced LMs:

  1. The linear mapping performance may come from feature encoding ability or any knowledge about recommendation (e.g., textual similarity or preference similarity). Random shuffle results are poor -> The performance largely comes from the knowledge about recommendation.

  2. The linear mapping performance is excellent -> Previous works in NLP [1, 2, 3, 4, 5] suggest user preference similarities may be encoded in advanced LMs.

  3. Textually dissimilar items gather in the recommendation space -> user preference similarities beyond textual similarities are encoded in advanced LMs.

- **Highlight the contribution of this paper.** We would like to highlight the two key contributions of this paper. First, we are the paper adopting the linear mapping method to study the knowledge about recommendation encoded in LMs, which has been widely used in the NLP community but rarely appears in the recommendation community. Second, we prove that huge potential exists in the representations from current advanced LMs. With simple model design, such representations yield superior performance across various tasks.

- **Verify the claim.** Following the suggestions of reviewer  $\color{orange}{\text{EzpW}}$, we add experiments on sequential recommendation and measure how the knowledge inside LM increases with the rise of LM size.

- **More baselines.** According to the suggestions of reviewer $\color{green}{\text{tJs7}}$, we add two more baselines DirectAU and GraphAU.

- **Comparing with LLM4Rec baselines.** According to the suggestions of reviewer $\color{orange}{\text{EzpW}}$, we compare the zero-shot performance of AlphaRec with LLM4Rec baseline LLMRank [6].

[1] Understanding intermediate layers using linear classifier probes. 2017 ICLR.

[2] Language models represent space and time. 2024 ICLR.

[3] Emergent world representations: Exploring a sequence model trained on asynthetic task. 2023 ICLR

[4] Probing pretrained language models for lexical semantics. 2020 EMNLP.

[5] A Structural Probe for Finding Syntax in Word Representations. 2019 NAACL.

[6] Large language models are zero-shot rankers for recommender systems. 2024 ECIR.

---

### Decision · Program_Chairs · 2024-09-25

**Decision:**

Reject

**Comment:**

The initial reviews were mixed, with ratings ranging from 2 to 7. Several of the reviewers' comments were minor (including comments regarding the method's name and prompt), and the authors provided reasonable justifications for them. The authors also provided additional results on a new dataset and with two new recent methods (although providing confidence intervals or another measure of variance would help, especially since some of the difference in performance seems relatively modest).

Other reviewer comments questioned the novelty of the approach. While the fact that item representations in LM space contain a strong recommendation signal is interesting, the reviewers note that this is perhaps a more interesting finding for recsys practitioners and less so for a more general ML audience. I don't fully agree since exploring LMs in specific domains interests the community and can lead to more generalizable insights. At the same time, the initial finding may warrant further analysis.

Reviewer EzpW and the authors had a productive discussion that led to several new results and convinced the reviewer to update their assessment (theirr rating went up from 2 to 4). Nonetheless, even after the discussion, the reviewer finds the paper requires a major re-write before it can be accepted to emphasize its contribution and recognize current literature that already makes use of language-based embeddings. The authors seem to broadly agree with this. The reviewer also suggests providing more analysis regarding LM representations and how adding item descriptions can make them even more accurate (there might also be a practical aspect here for new items). I would be curious to get the authors' hypothesis about how far these ideas will likely generalize to other domains where objective textual descriptions might be less revealing of preferences (perhaps music?).

I also acknowledge your message from Aug. 13, which I didn't get a chance to reply to before the deadline (the same day).

In our internal discussion, the reviewer re-expressed concerns about the technical novelty of the approach and the paper's current presentation. The seemingly most positive reviewer tJs7, recognized the concerns raised by other reviewers and did not object to the paper being rejected. Overall, this means that we were close to a consensus, even though I don't find the scores to reflect the quality of the paper.

I am sorry that I cannot recommend acceptance at this stage. I hope the reviewer's comments will help prepare the next version of this work.